# Transliteration: A Simple Technique For Improving Cross-Lingual Language Modeling

## Abstract

Impressive performance in natural language processing tasks has been achieved for many languages by transfer learning from large pretrained Multilingual Language Models (MLLM). However, unavailability of large corpora for most languages currently poses a significant hurdle. Thus it is important for MLLMs to extract the most out of existing corpora. In this regard, script diversity presents a challenge to MLLMs by reducing lexical overlap between closely related languages. Therefore, transliterating closely related languages that use different writing scripts to a common script may improve the downstream task performance of MLLMs. In this paper, we empirically measure the effect of transliteration on the performance of MLLMs by focusing on the Indo-Aryan language family, which has the highest script diversity. We pretrain two ALBERT models from scratch, where one is pretrained with original scripts and the other after transliterating to a common script. Afterward, we evaluate the models on the IndicGLUE benchmark. We perform Mann-Whitney U test on the performance metrics of the models to rigorously verify whether the effect of transliteration is significant or not. We find that transliteration benefits the low-resource languages without negatively affecting the comparatively high-resource languages. We also measure the cross-lingual representation similarity (CLRS) of the models using centered kernel alignment (CKA) on parallel sentences of eight languages from the FLORES-101 dataset. We find that the hidden representations of the transliteration-based model have higher and more stable CLRS scores.

## 1 Introduction

In the last few years, we have seen impressive advances in many NLP tasks. These advances have been primarily led by the availability of large representative corpora and improvement in the architecture of large language models. While improving model architectures, training methods, regularization techniques, etc., can help advance the state of NLP in general, the unavailability of large, diverse corpora is the bottleneck for most languages (Joshi et al., 2020). However, creating representative and inclusive corpora is an ongoing process and is not always possible for many low-resource languages. Thus to inclusively advance the state of NLP across languages, it is crucial to develop techniques for training MLLMs that can extract the most out of existing multilingual corpora.

Here we focus on the issue of diverse writing scripts used by closely related languages that may prevent MLLMs from learning good cross-lingual representations. For example, the Indo-Aryan family of languages uses at least six different scripts among them. Previous papers (Pfeiffer et al., 2021) have noted that low-resource languages that use unique scripts tend to have very few tokens representing them at the tokenizer. As a result, these languages tend to have more *UNK*nown tokens, and the words in these languages tend to be more split up by sub-word tokenizers. While this issue is evident for Indian languages, there are other examples of closely related languages separated by different scripts such as Serbian and Croatian. Often we can easily transliterate from one script to another using rule-based systems. For example, there are established standards that can be used to transliterate Greek (ISO 843), Cyrillic (ISO 9), Indic scripts (ISO 15919), and Thai (ISO

11940) to the Latin script. In this paper, We focus on the Indo-Aryan language family and empirically measure the effect of transliteration on the downstream performance of MLLMs. The linguistic relatedness in the Indo-Aryan language family is discussed in appendix A.1.

First, we pretrain two ALBERT base (Lan et al., 2020) models from scratch on the Indo-Aryan languages of the OSCAR corpus (Ortiz Su'arez et al., 2019), where one is pretrained with original scripts and the other after transliterating to a common script. Then, we evaluate the models on four diverse downstream tasks from the IndicGLUE benchmark dataset, which are Wikipedia Section Title Prediction (WSTP), News Category Classification (NCC), Named Entity Recognition (NER), and Cloze-Style Question Answering (CSQA). Additionally, we evaluate the models on five different public datasets spanning three different tasks, Article Genre Classification (AGC), Sentiment Analysis (SA), and Discourse Mode Classification (DMC). We use nine random seeds on all downstream finetuning tasks. To rigorously compare the two models, we perform the Mann-Whitney U test (MWU) between the uni-script model (group 1) and the multi-script model (group 2). MWU test lets us verify whether the performance differences of the two models are significant or not. Apart from statistical significance, we also report three different effect sizes to conclusively determine whether the magnitude of improvement due to transliteration is useful in practice. Using the MWU test, we conclude that transliteration significantly benefits the low-resource languages without negatively affecting the comparatively high-resource languages.

We also measure the tokenizer quality and the cross-lingual representation similarity (CLRS) to understand why the uni-script model performs better than the multi-script model. To measure tokenization quality, we use subword fertility and unbroken ratio (Ács, 2019; Rust et al., 2021). To measure the CLRS, we use the centered kernel alignment (CKA) (Kornblith et al., 2019) similarity score. We measure the CKA similarity score between the hidden representations of the models on the parallel sentences of eight Indo-Aryan languages from the FLORES-101 dataset(Goyal et al., 2021). We find that, compared to the multi-script model, the uni-script model achieves a higher CKA score and it is more stable throughout the hidden layers of the uni-script model. Based on this, we conclude that the uni-script model learns better cross-lingual representation than the multi-script model.

In summary, our contributions are primarily three-fold:

1. We find that transliteration significantly benefits the low-resource languages without negatively affecting the comparatively high-resource languages.

2. We establish this finding through rigorous experiments and show the statistical significance along with the effect size of transliteration using the Mann-Whitney U test.

3. Using CKA on the FLORES-101 dataset, we show that transliteration helps MLLMs learn better cross-lingual representation.

## 2 MOTIVATION AND BACKGROUND

In their study, Joshi et al. (2020) showed the resource disparity between low-resource and high-resource languages, and (Ruder, 2020) discussed the necessity of working with low-resource languages. A large body of work suggests that language-relatedness can help MLLMs achieve better performance on low-resource languages by leveraging related high-resource languages. For instance, (Pires et al., 2019) found that lexical overlap improved mBERT's multilingual representation capability even though it learned to capture multilingual representations with zero lexical overlaps. (Dabre et al., 2017) showed that transfer learning in the same or linguistically similar language family gives the best performance for NMT. (Lauscher et al., 2020) found that language relatedness is crucial for POS-tagging and dependency parsing tasks. Although, corpus size is much more important for tasks like NLI and Question Answering. (Wu & Dredze, 2020) demonstrated bilingual BERT can outperform monolingual BERT on low-resource languages when the bilingual languages are linguistically closely related. Nevertheless, mBERT outperforms bilingual BERT when it comes to low-resource languages. However, one of the major challenges in leveraging transfer between high-resource and low-resource languages is overcoming the script barrier. Script

barrier exists when multiple closely related languages use different scripts. (Anastasopoulos & Neubig, 2019) found that for morphological inflection, script barrier between closely related languages impedes cross-lingual learning. However, transliteration and phoneme-based techniques have been proposed to solve this issue. For example, (Murikinati et al., 2020) expanded upon (Anastasopoulos & Neubig, 2019) and showed that both transliteration and grapheme to phoneme (g2p) conversion removes script barrier and improves cross-lingual morphological inflection and (Rijhwani et al., 2019) showed that pivoting low-resource languages to their closely related high-resource languages results in better zero shot entity linking capacity and used phoneme-based pivoting to overcome the script barrier. (Bharadwaj et al., 2016) showed that phoneme representation outperformed orthographic representations for NER. (Chaudhary et al., 2018) also used phoneme representation to resolve script barrier and adapt word embeddings to low-resource languages. (Goyal et al., 2020) and (Song et al., 2020) both utilized transliteration and showed that language relatedness was required for improving performance on NMT. (Amrhein & Sennrich, 2020) studied how transliteration improved NMT and came to the conclusion that transliteration offered significant improvement for low-resource languages with different scripts. (Khemchandani et al., 2021) showed on Indo-Aryan languages that language relatedness could be exploited through transliteration along with bilingual lexicon-based pseudo-translation and aligned loss to incorporate low-resource languages into pretrained mBERT. (Muller et al., 2021a) showed that for unseen languages, script barrier hindered transfer between low-resource and high-resource languages for MLLMs and transliteration removed this barrier. They showed that transliterating Uyghur, Buryat, Erzya, Sorani, Meadow Mari, and Mingrelian to Latin script and finetuning mBERT on the respective corpus with masked language modeling objective improved their downstream POS performance significantly. (Dhamecha et al., 2021) showed that transliterating Indo-Aryan languages improved multilingual language model performance. In contrast, (K et al., 2020) and (Artetxe et al., 2020) seem to show that multilingual BERT can learn cross-lingual representations without any lexical overlap, a shared vocabulary, or joint training. However, these works focus on zero shot cross-lingual transfer learning where the models are finetuned on one language and evaluated on another.

From the literature, it can be seen that apart from g2p, many in the community believe transliteration to be a potential solution for script barriers. However, most of the work shows the benefits of transliteration for NMT. Nevertheless, there is no solid empirical analysis of the effects of transliteration for MLLMs apart from (Dhamecha et al., 2021; Muller et al., 2021a), which are our contemporary studies. Hence, the motivation behind this paper is to provide a solid empirical analysis of the effect of transliteration for MLLMs with statistical analysis and determine if it helps models learn better cross-lingual representation.

Several techniques have recently been used to study the hidden representations of multilingual language models. (Kudugunta et al., 2019) study CLRS of NMT models using SVCCA (Raghu et al., 2017). (Singh et al., 2019) used PWCCA (Morcos et al., 2018) to study the CLRS of mBERT and found that it drastically falls with depth. (Wu et al., 2020) have used CKA to study the CLRS of bilingual BERT models. They found that similarity is highest in the first few layers and drops moderately with depth. (Muller et al., 2021b) used CKA to study CLRS of mBERT before and after finetuning on downstream tasks. They found in all cases that CLRS increases steadily in the first five layers, then it decreases in the later layers. From this, they conclude that mBERT learns multilingual alignment in the early layers and preserves it throughout finetuning. A contemporary work (Del & Fishel, 2021) has applied various similarity measures to understand CLRS of various multilingual masked language models. Their results also show that CLRS increases in the first half of the models, while in the later layers, this similarity steadily falls.

## 3 Experiment and Results

### 3.1 Dataset

We pretrained two ALBERT base models from scratch on a subset of the OSCAR corpus. OSCAR corpus is a large multilingual corpus extracted from Common Crawl that contains hundreds of different languages. We use the unshuffled deduplicated version of OSCAR

corpus available via Huggingface datasets library (Lhoest et al., 2021). We pretrain on Panjabi, Hindi, Bangla, Oriya, Assamese, Gujarati, Marathi, Sinhala, Nepali, Sanskrit, Goan Konkani, Maithili, Bihari, and Bishnupriya portion of the OSCAR corpus. The pretraining dataset details are given in appendix A.5.

We evaluate the models on four downstream tasks from IndicGLUE, which are NCC, WSTP, CSQA, and NER (Kakwani et al., 2020). In addition, we evaluate the models on other publicly available datasets. Specifically, on BBC Hindi News Classification, Soham Bangla News Classification, iNLTK Headlines Classification, IITP Movie, and Product Review Sentiment Analysis(Akhtar et al., 2016), and MIDAS Discourse Mode Classification(Dhanwal et al., 2020) datasets.

## 3.2 Pretraining

**Corpus Preparation:** Since the OSCAR corpus contains raw text from the Web, we apply a few filtering and normalization. First, we discard entries where the dominant script does not match the language tag provided by the OSCAR corpus. Then we use the IndicNLP normalizer(Kunchukuttan, 2020) to normalize the raw text. For XLM-Indic, we then transliterate all the text to ISO-15919 format using the Aksharamukha(Rajan, 2015) library.

**Tokenizer Training:** Next, we train two SentencePiece tokenizers (Kudo & Richardson, 2018) on the transliterated and the non-transliterated corpus with a vocabulary size of 50,000. Then we use the trained tokenizer and the sentence-splitter from the IndicNLP library to split long entries from the corpus at sentence boundaries so that no entry may have more than 512 tokens. Finally, we discard short entries (<512 characters) to improve the training efficiency.

**ALBERT Model Training:** We first pretrained an ALBERT base model from scratch on the non-transliterated corpus as our baseline. Afterward, we pretrained another ALBERT base from scratch on the transliterated corpus. We chose the ALBERT base model due to computing constraints. We trained the models on a single TPUv3 VM. Both models were trained using the same hyperparameters. We followed the hyperparameters used in (Lan et al., 2020) except for batch size and learning rate. The pretraining objective is also the same ase (Lan et al., 2020). The hyperparameter values and details are presented in the appendix A.3. Each model took about 7.5 days to train. We use the ALBERT implementation from the Huggingface Transformers Library (Wolf et al., 2020). Henceforth, unless mentioned otherwise, the non-transliteration or multi-script model will be called the baseline model, and the transliteration or uni-script model will be called XLM-Indic.

## 3.3 Statistical Analysis

We perform statistical analysis to determine if the performance differences between baseline and XLM-Indic are significant. In short, the statistical analysis tells us the effect of transliteration on model performance. For this purpose, we perform Mann–Whitney U test (MWU) (Mann & Whitney, 1947; Wilcoxon, 1945). MWU is a non-parametric hypothesis test between two groups/populations. MWU is chosen because it has weak assumptions. The only assumptions of MWU are that the samples of the two groups are independent of each other, and the samples are ordinal. Under the MWU, our null hypothesis or $\mathbf{h_0}$ is that the performances of XLM-Indic (group 1) and baseline (group 2) are equivalent, and the alternative hypothesis or $\mathbf{h_a}$ is that the performances (groups) are non-equal. We set our confidence interval $\alpha$ at 0.05 and reject the $\mathbf{h_0}$ for p-values $< \alpha$. We also report three test statistics as p-value only give statistical significance, which can be misleading at times(Sullivan & Feinn, 2012).

The test statistics are three different effect sizes that convey three different information. These test statistics are absolute effect size ($\delta$), common language effect size ($\rho$ ), and standardized effect size. The absolute effect size $\delta$ is the difference between the mean of the models' performance metric, which is given as,

$$\delta = \mu_{\text{XLM-Indic}} - \mu_{\text{Baseline}}$$

for any given task and language. For any given task, if the $\mathbf{h_0}$ is rejected, a positive $\delta$ indicates XLM-Indic is better, and a negative $\delta$ indicates the baseline is better. $\rho$ gives us the probability of superiority of one group being better performing between given two groups. That is the probability that a random performance sample of XLM-Indic is greater than a random performance sample of the baseline. The last test statistics is standardized effect size which indicates the magnitude of difference between the performance values of XLM-Indic (group 1) and baseline (group 2). Standardized effect size shows us how realistically significant the performance differences are between models even if the performance difference is statistically significant. It gives us a value between 0 to 1. and its ranges are: **small effect** ( $0 \leq$ std. effect $\leq 0.3$) , **medium effect** ( $0.3 <$ std. effect $\leq 0.5$) and **large effect** ($0.5 <$ std. effect). We show the standardized effect size only on public datasets. We performed MWU on all downstream tasks except CSQA. On CSQA, we only report the $\delta$. The MWU is performed using the SciPy library (Virtanen et al., 2020), and the results are further validated using (2017, 2017).

## 3.4 Downstream Finetuning

We finetune the models on each downstream task independently. The hyperparameters are selected based on the recommendations in (Mosbach et al., 2021). The specific hyperparameters used for each task are reported in the appendix A.4. On all tasks, we finetune with nine random seeds and report the average and standard deviation of the metrics. In Table 1 and Table 3, we report the performances on IndicGLUE benchmark tasks and in Table 2 on other publicly available datasets. Here we discuss each of the tasks and compare the performance of XLM-Indic with the baseline model. We show the results from IndicBERT(Kakwani et al., 2020) and (Dhamecha et al., 2021) as independent baselines.

On some tasks, the performance of XLM-Indic is observably better than the baseline. For WSTP and NER, XLM-Indic shows improvement by several percentage points over the baseline on all languages. Conversely, on other tasks, the difference is nuanced, and it is not clear whether XLM-Indic is actually performing better or not. An example of this can be seen in the iNLTK Gujarati News Classification task shown in Table 2. We discuss these nuanced results in-depth in the latter part of this section under our statistical analysis.

**Wikipedia Section Title Prediction** This is a multiple-choice question-answer task from IndicGLUE. On this task, we predict the title of a Wikipedia article section from four choices. However, we train a sequence classification model instead of a multiple-choice question-answer model for simplicity. This task is the same across all languages. Thus we simultaneously finetune the model on all these languages. In Table 1, we report the test set accuracy of different models on this task. Here, XLM-Indic performs significantly better than the baseline. A $\rho$ value of 1 for all languages indicates that XLM-Indic will always perform better than the baseline. The $\delta$ value of Oriya (7.12), Gujarati (6.02), and Assamese (5.04) indicates that transliteration helps comparatively low-resource languages more compared to Hindi (4.06) and Bengali (3.02).

**News Category Classification:** This is a collection of news article classification tasks from the IndicGLUE benchmark on five languages. The dataset sizes vary widely. For example, the Oriya dataset is about ten times the Gujarati dataset. Thus this task helps us test the finetuning capability of the models on classification tasks with different dataset sizes. In Table 1, we report the test set accuracy of different models on this task. We see our first failure to reject the null hypothesis case for the Bengali and Marathi classification tasks. Even though the $\delta$ is negative, we can not reject the null hypothesis for Bengali and Marathi as the p-values are 0.181 and 0.1683, respectively. The $\rho$ for these two tasks show that XLM-Indic can outperform the baseline only 31% of the time. For Panjabi ($\delta$ of 1.07) and Oriya ($\delta$ of 0.68), the $\rho$, indicates that XLM-Indic can always outperform baseline. However, for Gujarati ($\delta$ of 0.60), XLM-Indic can outperform the baseline 80% of the time.

Table 1: Results on Classification Tasks from IndicGLUE Benchmark

| Model | pa | hi | bn | or | as | gu | mr | avg |
|---|---|---|---|---|---|---|---|---|
| **Wikipedia Section Title Prediction** | | | | | | | | |
| **Independent Baseline** | | | | | | | | |
| mBERT (Dhamecha et al., 2021) | 76.48 | 80.81 | 82.85 | 28.29 | - | 78.58 | 84.38 | 71.90 |
| XLM-R (Kakwani et al., 2020) | 70.29 | 76.92 | 80.91 | 68.25 | 56.96 | 27.39 | 77.44 | 65.45 |
| IndicBERT base (Kakwani et al., 2020) | 67.39 | 74.02 | 80.11 | 57.14 | 65.82 | 68.79 | 72.56 | 69.40 |
| IndicBERT large (Kakwani et al., 2020) | 77.54 | 77.80 | 82.66 | 68.25 | 56.96 | 52.23 | 77.44 | 70.41 |
| **Ours** | | | | | | | | |
| Baseline | 74.33±0.83 | 78.18±0.33 | 81.18±0.28 | 74.35±1.2 | 76.70±0.83 | 76.37±0.53 | 79.10±0.84 | 77.17 |
| XLM-Indic | 77.55±0.61 | 82.24±0.18 | 84.38±0.29 | 81.47±0.99 | 81.74±0.82 | 82.39±0.27 | 82.74±0.52 | 81.78 |
| **Test statistics** | | | | | | | | |
| $\delta$ | **3.22** | **4.06** | **3.20** | **7.12** | **5.04** | **6.02** | **3.64** | **4.61** |
| p-value | 0.0004 | 0.0004 | 0.0004 | 0.0004 | 0.0004 | 0.0004 | 0.0004 | - |
| $\rho$ | 1 | 1 | 1 | 1 | 1 | 1 | 1 | - |
| **News Category Classification** | | | | | | | | |
| **Independent Baseline** | | | | | | | | |
| XLM-R (Kakwani et al., 2020) | 94.87 | - | 98.29 | 97.07 | - | 96.15 | 96.67 | 96.61 |
| mBERT (Kakwani et al., 2020) | 94.87 | - | 97.71 | 69.33 | - | 84.62 | 96.67 | 88.64 |
| IndicBERT base (Kakwani et al., 2020) | 97.44 | - | 97.14 | 97.33 | - | 100.00 | 96.67 | 97.72 |
| IndicBERT large (Kakwani et al., 2020) | 94.87 | - | 97.71 | 97.60 | - | 73.08 | 95.00 | 91.65 |
| **Ours** | | | | | | | | |
| Baseline | 96.83±0.19 | - | 98.14±0.14 | 98.09±0.16 | - | 98.80±0.43 | 99.58±0.25 | 98.30 |
| XLM-Indic | 97.90±0.17 | - | 97.99±0.22 | 98.77±0.12 | - | 99.40±0.54 | 99.47±0.21 | 98.70 |
| **Test statistics** | | | | | | | | |
| $\delta$ | **1.07** | - | **-0.15** | **0.68** | - | **0.60** | **-0.18** | **0.40** |
| p-value | 0.0003 | - | 0.181 | 0.0004 | - | 0.03084 | 0.1683 | - |
| $\rho$ | 1 | - | 0.31 | 1 | - | 0.80 | 0.31 | - |
| **NER (F1-Score)** | | | | | | | | |
| **Independent Baseline** | | | | | | | | |
| mBERT (Dhamecha et al., 2021) | 50.00 | 86.56 | 91.81 | 19.05 | 92.31 | 68.04 | 91.27 | 71.29 |
| XLM-R (Kakwani et al., 2020) | 17.86 | 89.62 | 92.95 | 25.00 | 66.67 | 55.32 | 87.86 | 62.18 |
| IndicBERT base (Kakwani et al., 2020) | 21.43 | 90.30 | 93.39 | 8.69 | 41.67 | 54.74 | 88.71 | 56.69 |
| IndicBERT large (Kakwani et al., 2020) | 44.44 | 86.81 | 91.85 | 35.09 | 43.48 | 70.21 | 87.73 | 65.66 |
| **Ours** | | | | | | | | |
| Baseline | 76.69±1.5 | 91.80±0.42 | 96.39±0.19 | 84.18±1.8 | 75.45±1.8 | 69.10±2.9 | 88.72±0.40 | 83.19 |
| XLM-Indic | 85.42±1.9 | 92.93±0.21 | 97.31±0.22 | 93.54±0.58 | 89.06±2.2 | 80.16±0.15 | 90.56±0.44 | 89.85 |
| **Test statistics** | | | | | | | | |
| $\delta$ | **8.73** | **1.13** | **0.92** | **9.36** | **13.61** | **11.06** | **1.84** | **6.66** |
| p-value | 0.0004066 | 0.0004066 | 0.0003983 | 0.0004038 | 0.000401 | 0.0004066 | 0.0004095 | - |
| $\rho$ | 1 | 1 | 1 | 1 | 1 | 1 | 1 | - |

orange indicates baseline and XLM-Indic are equal and blue indicates XLM-Indic is better

**Named Entity Recognition:** We use the balanced Wikiann dataset from Rahimi et al. (2019) for this task. Similar to WSTP, we simultaneously finetune the models on all languages. In Table 1, we report the F1-scores of different models. As per the p-value, XLM-Indic's improvement over baseline is statistically significant. We specifically see that as per $\delta$, XLM-Indic makes significant improvements on Panjabi, Oriya, Assamese, and Gujarati NER. The $\rho$ for all languages indicate that XLM-Indic can always outperform the baseline on all Indo-Aryan family of languages. Similar to WSTP, the $\delta$ values show that transliteration helps low-resource languages more. For instance, the $\delta$ for Panjabi, Oriya, Assamese, and Gujarati are 8.73, 9.36, 13.61, and 11.06, respectively. We can also see improvement on Hindi (1.13), Bengali (0.92), and Marathi (1.84) NER.

Table 2: Results on Public Datasets

| Language | Dataset | IndicBERT-base | Baseline | | XLM-Indic | | $\delta$ | p-value | $\rho$ | Test Statistics Standardized effect size F1-score |
|---|---|---|---|---|---|---|---|---|---|---|
| | | | Accuracy | F1-score | Accuracy | F1-Score | | | | |
| **Article Genre Classification** | | | | | | | | | | |
| hi | BBC News | 74.60 | 77.28±1.5 | 46.47±4.6 | 79.14±0.60 | 48.59±0.27 | **2.12** | 0.4363 | 0.62 | 0.19 |
| bn | Soham News Article Classification | 78.45 | 93.22±0.49 | 91.12±0.85 | 93.89±48 | 91.75±0.73 | **0.63** | 0.05031 | 0.78 | 0.46 |
| gu | INLTK Headlines | 92.91 | 90.41±0.69 | 88.82±0.73 | 90.73±0.75 | 89.14±0.88 | **0.32** | 0.5457 | 0.59 | 0.15 |
| mr | INLTK Headlines | 94.30 | 92.21±0.23 | 89.23±0.54 | 92.04±0.47 | 89.09±0.78 | **-0.14** | 0.6665 | 0.43 | 0.10 |
| **Sentiment Analysis** | | | | | | | | | | |
| hi | IITP Product Reviews | 71.32 | 76.33±0.84 | 74.04±0.99 | 77.18±0.77 | 74.53±0.98 | **0.49** | 0.3865 | 0.63 | 0.21 |
| hi | IITP Movie Reviews | 59.03 | 65.91±2.2 | 65.26±2.2 | 66.34±0.16 | 65.87±1.6 | **0.61** | 0.5457 | 0.59 | 0.15 |
| **Discourse Mode Classification** | | | | | | | | | | |
| hi | MIDAS Discourse | 78.44 | 78.39±0.33 | 47.26±6.2 | 78.54±0.91 | 63.80±3.2 | **16.54** | 0.00004 | 1 | 0.83 |

orange indicates baseline and XLM-Indic are equal and blue indicates XLM-Indic is better

**Article Genre, Sentiment & Discourse Mode Classification:** We evaluate the models on publicly available sequence classification datasets. Most of these classification tasks are highly skewed. Thus we report F1 scores in addition to accuracy on these tasks. We encourage others to do so in the future. As stated earlier, apart from $\delta$ and $\rho$, we also report standardized effect size on these datasets as their difference were not as pronounced as the IndicGLUE tasks. **These tasks show one of the major reasons why multi-seeded results along statistical analysis are imperative**. We present our results in Table 2. We only show the test statistics for the F1-score here. The test statistics for accuracy are provided in appendix A.6. For BBC News classification, we can see that even though the $\delta$ for Hindi is 2.12, there is no statistically significant difference in the models' performance. The null hypothesis can not be rejected due to the p-value being 0.4363, and the standardized effect size is small (0.19). Both of these are due to the standard deviation of the baseline model, which is ($\pm 4.6$). However, $\rho$ indicates that XLM-Indic would outperform the baseline 62% of the time. The Soham Articles classification, a Bengali news classification task with six classes, is marginally rejected due to the p-value being 0.05031. Nevertheless, the standardized effect size is medium (0.46), and as per $\rho$, XLM-Indic outperforms the baseline 78% of the time. For INLTK Headline classification task, we see that both models are equivalent for Gujarati and Marathi with a p-value of 0.5457 and 0.6665, respectively. Both of them have a small effect size (Gujarati 0.15 and Marathi 0.10). The $\rho$ indicates that XLM-Indic outperforms the baseline 59% of times for Gujarati and 43% of times for Marathi. On sentiment analysis tasks, we see a similar trend. Both models are equivalent performance wise with a p-value of 0.3865 for IITP Product Review and 0.5457 for IITP Movie review. The standardized effect size for the tasks is 0.21 and 0.15, respectively, which indicates that the standardized effect is small for both tasks. Apart from that, as per $\rho$, XLM-Indic outperforms the baseline 63% of times for IITP Product review and 59% of the time for IITP Movie review. Lastly, for Discourse Mode Classification, we see that XLM-Indic outperforms the baseline with a $\delta$ value of 16.54 and a p-value of 0.00004. The standardized effect is large (0.83) and $\rho$ indicates that XLM-Indic will always outperform the baseline.

**Effect of Transliteration:** In general, we can see that the effect of transliteration is mostly positive. However, it does not hurt the model's performance either, as we can see that the models performed equivalently in all the tasks where the null hypothesis could not be rejected. Transliteration mostly helps the comparatively low-resource languages (Panjabi, Oriya, Assamese), while high-resource languages also (Hindi, Bengali) see good improvements as seen in WSTP and NER tasks. As for transliteration not hurting model performance, we can see that in Table 2 excluding Discourse Mode Classification, in all other tasks, XLM-Indic performance was equivalent to the baseline. While Discourse Mode Classification had a statistically significant performance difference.

## 3.5 Zero Shot Capability Testing

We use the CSQA task to test the zero shot capability of the models as we can use the pretrained masked language models without finetuning on this task. On CSQA, we are given four entities to choose from to fill in a masked entity in a sentence. This task is designed to test whether language models can be used as knowledge bases (Petroni et al., 2019). In Table 3 we report the results. We see that XLM-Indic performs better on all languages. The $\delta$ is particularly impressive in Gujarati (10.36), Assamese (3.44), and Marathi (3.39). This indicates that XLM-Indic is better at representing entity knowledge than the baseline model. This is not surprising as many entity names have the same spelling regardless of the script used. XLM-Indic maps these to the same token, which helps it to construct better knowledge representation. We found some issue with IndicBERT evaluation code for this task. We discuss this issue in appendix A.2. We evaluate IndicBERT with our code and report the results in Table 3.

Table 3: Test accuracy on CSQA

| Model | pa | hi | bn | or | as | gu | mr | avg |
|---|---|---|---|---|---|---|---|---|
| **Cloze-style QA (Zero Shot)** | | | | | | | | |
| **Independent Baseline** | | | | | | | | |
| IndicBERT base (Our Evaluation) | 29.33 | 30.76 | 28.45 | 30.38 | 29.98 | 81.50 | 29.71 | 37.16 |
| | | | | | | | | |
| **Ours** | | | | | | | | |
| Baseline | 31.04 | 36.72 | 35.19 | 34.63 | 33.92 | 59.86 | 36.14 | 38.21 |
| XLM-Indic | 32.77 | 38.52 | 36.38 | 36.00 | 37.36 | 70.22 | 39.53 | 41.54 |
| δ | **1.73** | **1.8** | **1.19** | **1.37** | **3.44** | **10.36** | **3.39** | **3.33** |

orange indicates baseline and XLM-Indic are equal and blue indicates XLM-Indic is better than baseline

## 4 WHY TRANSLITERATION WORKS

In this section, we analyze why XLM-Indic performs better than the baseline model from two perspectives, tokenization quality, and CLRS.

### 4.1 TOKENIZER QUALITY ANALYSIS

As discussed in 1 and 2, we expect the transliteration model to exploit better tokenization across the languages. Whereas the same word written in six different scripts gets mapped to six different tokens in the baseline model, these get mapped to the same token in the transliteration model. Thus we expect the tokenizer to more efficiently tokenize a given text in the case of the transliteration model.. Following (Ács, 2019) and (Rust et al., 2021),

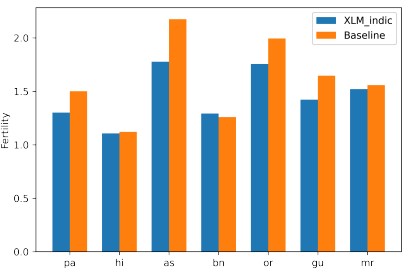
(a) Subword Fertility.

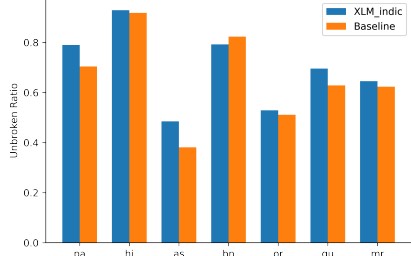
(b) Unbroken Ratio.

Figure 1: Subword fertility (lower is better) and unbroken ratio (higher is better)

we measure the subword fertility (average number of tokens per word) and the ratio of words unbroken by the tokenizer. From figure 1, we can see that transliteration reduces the splitting of words. This indicates that many words that were represented by different tokens in the baseline model are represented by a single token in the transliteration model.

### 4.2 CROSS-LINGUAL REPRESENTATION SIMILARITY

Following (Muller et al., 2021b), (Wu et al., 2020) and (Del & Fishel, 2021) we apply CKA to measure CLRS. We use the CKA implementation from the Ecco library (Alammar, 2021). We use parallel sentences on eight Indo-Aryan languages from the FLORES-101 (Goyal et al., 2021) dataset. This is a dataset of professionally translated parallel sentences on 101 languages. We only consider Panjabi, Hindi, Bengali, Oriya, Assamese, Gujarati, Marathi, and Nepali sentences from the FLORES-101 dataset as these are the only ones present in the pretraining corpus.

First, we extract the hidden state representations of the models on these parallel sentences. Then for each language pair, we calculate the CKA similarity score between the hidden state representations of the corresponding layers. For example, we calculate the CKA similarity

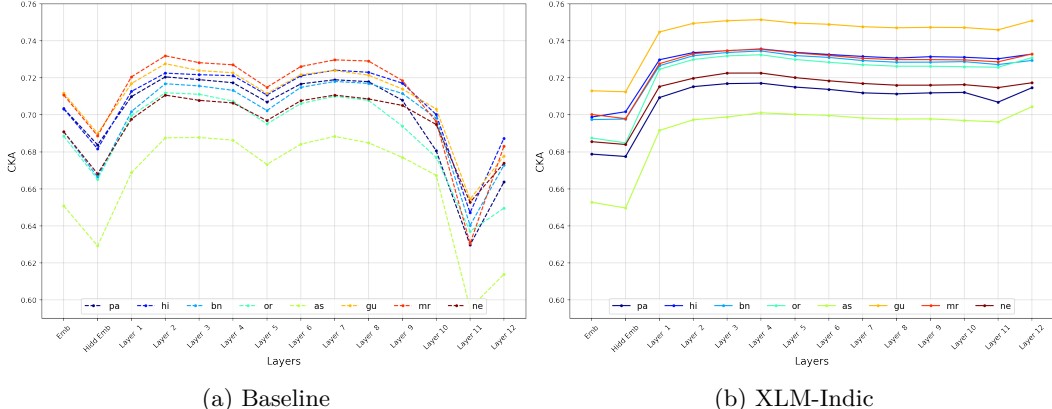

(a) Baseline                       (b) XLM-Indic

Figure 2: CKA Similarity Score for the baseline model and XLM-Indic

score between layer eight hidden state representations of Bengali and Hindi parallel sentences. Then for each language, we average its CKA similarity scores with other languages per layer. In Figure 2 we plot this average CKA similarity for each layer of the models for each language. We see that XLM-Indic retains high CLRS score throughout the model. On the other hand, the CLRS score drops in the middle and end layers of the baseline model. Overall the CLRS score of XLM-Indic is more stable. This indicates that XLM-Indic has learned better cross-lingual representations. We show the individual similarity figures for all language pairs in appendix A.9.

## 5   Conclusion and Future Work

In this paper, we found that transliterating closely related languages to a common script improves MLLM performance and leads to learning better CLRS. We established this by conducting rigorous empirical and statistical analysis to quantify the statistical significance and effect size of transliteration on MLLM for Indo-Aryan languages which have the largest script diversity. We found that transliteration especially improved the performance of comparatively low-resource languages and did not hurt the performance of high-resource languages. This findings are in agreement with contemporary literature (Dhamecha et al., 2021; Muller et al., 2021a). Our results indicate that in other scenarios where closely related languages from a language family use different scripts, transliteration can be used to improve MLLM performance. For example, the Dravidian, South Slavic, and Turkic Family of languages present similar scenarios. We would also like to implore our peers to include more hypothesis tests in their studies against a strong baseline. Non-parametric tests like MWU are easy to use and have a high level of interpretability. In future, we would like to measure the impact of transliteration on larger models and more languages. Also, character-based models hold more potential to exploit the lexical overlap enabled by transliteration. Efficient Transformer models that can efficiently handle long sequences can be used to train these character-level models.

## 6   Reproducibility Statement

We have taken great care to ensure our results are reproducible. The dataset details are provided in subsection 3.1, and the data processing steps and pretraining details are described in 3.2. The MWU implementation details are provided in 3.3. Finetuning hyperparameters are provided in appendix A.4. We have also provided our code as supplementary material.

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

## A  Appendix

### A.1  Linguistic relatedness in Indo-Aryan family

Many South Asian and Southeast Asian languages are intimately connected linguistically, historically, phonologically(Littell et al., 2017) and phylogenetically. However, due to dif­ferent scripts, it is difficult for MLLMs to fully exploit this shared information. Among the languages we considered in this study we encounter six different scripts. These are shown in Table 5. Nevertheless, these scripts have shared ancestry from ancient Brahmic script (Hockett et al., 1997; Coningham et al., 1996) and have similar structures that we can easily use to transliterate them to a common script. The ISO-15919 standard has been designed to perform this transliteration. Also, many of these languages heavily borrow from Sanskrit, and due to its influence, many words are shared among these languages. Therefore, due to their relatedness and highly diverse script barrier, this family of languages presents a unique opportunity to analyze the effects of transliteration on MLLMs.

### A.2  Cloze Style QA IndicBERT Evaluation Issue

Since a word can be tokenized to multiple tokens by the subword tokenizer, correctly eval­uating the model on this task requires special care. Specifically, we have to use the same number of mask tokens as the number of subword tokens that a word gets split into. Then we calculate the probability for the word by multiplying the probability of the subword to­kens predicted by the masked language model. We found that on the IndicBERT evaluation code, only a single mask token was used irrespective of the number of subword tokens that a word gets split into. We do not think this is the correct way to evaluate a masked language model on this task.

### A.3 Pretraining Hyperparameters

We used a batch size of 256, which is the highest that fits into TPU memory, whereas the ALBERT paper used a batch size of 4096. As our batch size is $1/16^{\text{th}}$ of the ALBERT paper, we use a learning rate of 1e-3/8, which is approximately $1/16^{\text{th}}$ of the learning rate used in the ALBERT paper (1.76e-2). Additionally, we use the Adam optimizer(Kingma & Ba, 2015) instead of the LAMB optimizer. The rest of the hyperparameters were the same as the ALBERT paper. Specifically, we use a sequence length of 512 with absolute positional encoding, weight decay of 1e-2, warmup steps of 5000, max gradient norm of 1.0, and Adam epsilon of 1e-6. The models were trained for 1M steps.

### A.4 Downstream Hyperparameters

All of our hyperparameters for downstream tasks are presented in Table 4. The batch size was chosen to be the maximum that fits in memory. This was done so that each batch contains approximately the same number of tokens. Otherwise the hyperparameters were chosen following the recommendations of (Mosbach et al., 2021). On the highly IITP Movie Review, IITP Product Review and MIDAS Discourse we found that this default setting resulted in worse performance compared to the independent baselines. So we finetuned the learning rate and classifier dropout on the validation set of these tasks.

Table 4: Hyperparameters for all tasks

| Task | TPU | Batch Size | Learning Rate | Weight Decay | Dropout | Epochs | Warmup Ratio |
|---|---|---|---|---|---|---|---|
| News Category Classification | False | 16 | 2e-5 | 0.01 | 0.1 | 20 | 0.10 |
| Wikipedia Section-Title Prediction | True | 256 | 2e-5 | 0.01 | 0.1 | 3 | 0.10 |
| Named Entity Recognition | True | 512 | 2e-5 | 0.01 | 0.1 | 20 | 0.10 |
| BBC Hindi News Classification | False | 16 | 2e-5 | 0.01 | 0.1 | 20 | 0.10 |
| Soham Bangla News Classification | False | 16 | 2e-5 | 0.01 | 0.1 | 8 | 0.10 |
| iNLTK Headlines Classification | False | 256 | 2e-5 | 0.01 | 0.1 | 20 | 0.10 |
| IITP Movie Review | False | 64 | 5e-5 | 0.01 | 0.25 | 20 | 0.10 |
| IITP Product Review | False | 16 | 5e-5 | 0.01 | 0.5 | 20 | 0.10 |
| MIDAS Discourse Mode | False | 32 | 2e-5 | 0.01 | 0.5 | 20 | 0.10 |

### A.5 Dataset Details

Here we provide the corpus size details.

Table 5: Pretraining dataset details

| Language | Pretraining corpus size in GB | Language Sub-family | Script |
|---|---|---|---|
| hi | 8.9 | Central Indo-Aryan | Devanagari |
| bn | 5.8 | Eastern Indo-Aryan | Bengali-Assamese |
| mr | 1.4 | Southern Indo-Aryan | Devanagari |
| ne | 1.2 | Northern Indo-Aryan | Devanagari |
| si | 0.783 | Insular Indo-Aryan | Sinhala |
| gu | 0.705 | Western Indo-Aryan | Gujarati |
| pa | 0.449 | Northwestern Indo-Aryan | Gurmukhi |
| or | 0.18 | Eastern Indo-Aryan | Oriya |
| as | 0.069 | Eastern Indo-Aryan | Bengali-Assamese |
| sa | 0.036 | Sanskrit | Devanagari |
| bpy | 0.0017 | Eastern Indo-Aryan | Bengali-Assamese |
| gom | 0.0017 | Southern Indo-Aryan | Devanagari |
| bh | 0.000034 | Eastern Indo-Aryan | Devanagari |
| mai | 0.000011 | Eastern Indo-Aryan | Devanagari |

### A.6 Public datasets accuracy test statistics

The MWU p-values and test statistics for public datasets accuracy is given in Table 6. We can see that for BBC News (p-value 0.0088) and Soham Articles Classification (p-value 0.0090) XLM-Indic is better than the baseline with a $\delta$ of 1.86 and 0.67, respectively. Both tasks have large standardized effect size (0.62 for both). However, as per $\rho$, XLM-Indic outperforms the baseline 87% of the time. Whereas, the $\rho$ for Soham News Article Classification is 0.57. As for INLTK Headlines, XLM-Indic and the baseline perform equivalently. On INLTK Headlines, the p-value for Gujarati ($\delta$ of 0.32) is 0.6249 and Marathi ($\delta$ of -0.17) is 0.3503. On IITP Product Reviews, XLM-Indic outperforms the baseline with a $\delta$ of 0.85, p-value of 0.4099 and $\rho$ of 0.79. However, the standardized effect size is medium (0.48) for the task. In contrast, on IITP Movie Reviews, both models are equivalent performance wise with a $\delta$ of 0.15, p-value of 0.8941, $\rho$ of 0.52 and a small (0.031) standardized effect. Finally, we can see that both models performing equally on Discourse Mode Classification. The $\delta$ is 0.15 with a p-value of 0.7561 and a small (0.073) standardized effect size. However, as per the $\rho$, XLM-Indic outperforms the baseline 45% of the time.

Table 6: Public datasets test statistics of accuracy

| Language | Dataset | | Test Statistics | | |
|---|---|---|---|---|---|
| | | $\delta$ | p-value | $\rho$ | Standardized effect size |
| **Article Genre Classification** | | | | | |
| hi | BBC News | **1.86** | 0.0088 | 0.87 | 0.62 |
| bn | Soham News Article Classification | **0.67** | 0.0090 | 0.57 | 0.62 |
| gu | INLTK Headlines | **0.32** | 0.6249 | 0.57 | 0.12 |
| mr | INLTK Headlines | **-0.17** | 0.3503 | 0.36 | 0.22 |
| **Sentiment Analysis** | | | | | |
| hi | IITP Product Reviews | **0.85** | 0.04099 | 0.79 | 0.48 |
| hi | IITP Movie Reviews | **0.15** | 0.8941 | 0.52 | 0.031 |
| **Discourse Mode Classification** | | | | | |
| hi | MIDAS Discourse | **0.15** | 0.7561 | 0.45 | 0.073 |

orange indicates baseline and XLM-Indic are equal and blue indicates XLM-Indic is better

### A.7 Case Study on Soham Bangla News Classification:

In order to better understand the contrast between classification decision of XLM-Indic and baseline model, using the Layer Integrated Gradients method (Sundararajan et al., 2017) as a local self-explaining method, we performed multiple out of distribution adversarial examples to check the input features attribution difference between XLM-Indic and the baseline. The experiments were performed using Soham Bangla News Classification finetuned models. Soham Bangla News Classification was chosen due to it being a multi-class classification task. It has six classes which are 'entertainment', 'international', 'kolkata', 'national', 'sport' and 'state'. The adversarial examples were created by mixing class dependant words. For example, "They won the award for their acting performance." should be classified as entertainment. However, "They won the penalty by acting injured." should be classified as sports instead of entertainment. In our experiments, XLM-Indic tended to perform better compared to baseline and capture more correct class dependant words instead of the adversarial ones. However, providing certain class dependant keyword (i.e. 'police', 'political, 'actor', 'footballer') let the baseline model fix its missclassification. We provide one such example below.

We used the example, "সে রাস্তার পাশে উন্মাদের অভিনয় করতো কিন্তু রাত হলে সুযোগ পেলেই ডাকাতি করতো।", which translates to, "He used to act insane on the side of the road; however, he would

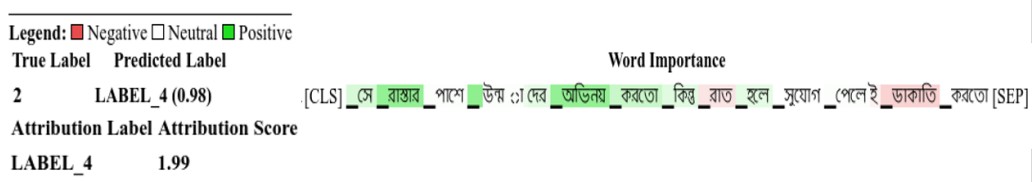

Figure 3: Feature importance for baseline

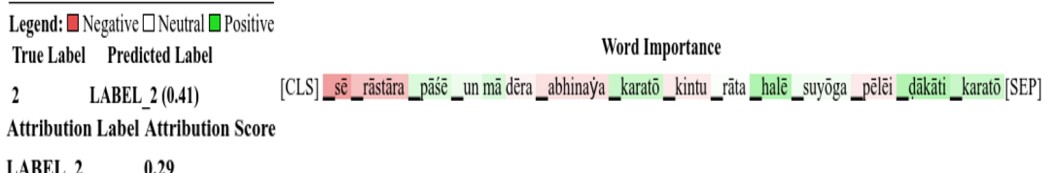

Figure 4: Feature importance for XLM-Indic

rob whenever he got the chance at night.". The true label for this example under Soham News Classification should be 'state'. Which is label 2 in the dataset. Here act is used as a class dependant adversarial word to deviate the models towards 'entertainment'. Whereas, we would like the models to attribute 'rob', which is the the correct word . Figure 4 shows the word attributions for the baseline model. We can see that the baseline classified the sentence as 'entertainment' (label 4) and put positive attribution on অভিনয় , which means act. On the other hand it put negative attribution on ডাকাতি, which means rob. However, XLM-Indic correctly classified the sentence and puts the expected positive and negative attributions on 'rob' and 'act' respectively. Nevertheless, as stated earlier, allowing certain class dependant keywords fixes the baselines misclassification. For example, we found that adding 'Police report states that' in front of our provided example fixed the baseline models prediction to the correct class.

## A.8 PRETRAINING LOSS CURVES

Our pretraining loss curve is provided in figure 5

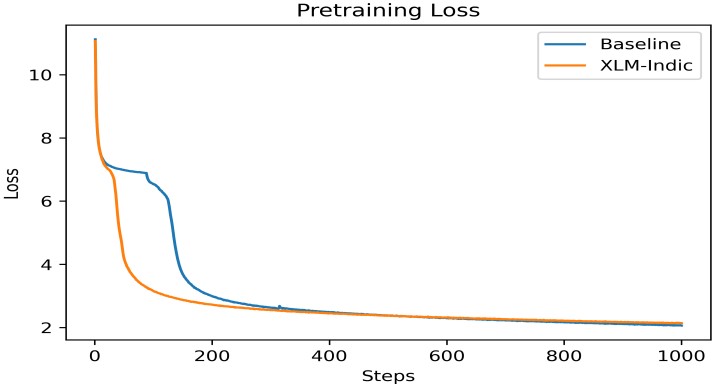

Figure 5: Pretraining loss: Baseline vs Transliteration model

## A.9 CROSS-LINGUAL SIMILARITY OF ALL LANGUAGE PAIRS

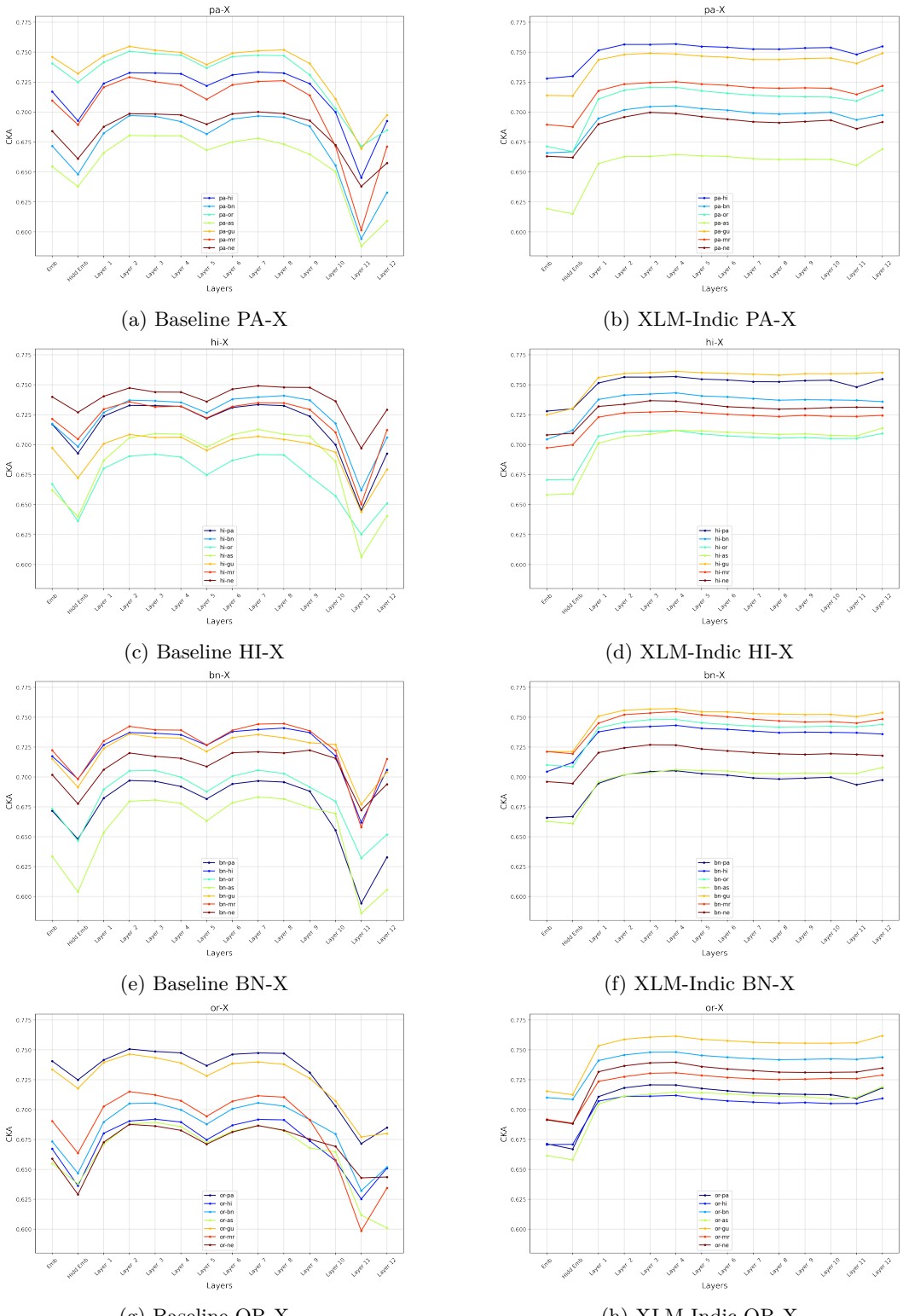

(a) Baseline PA-X      (b) XLM-Indic PA-X

(c) Baseline HI-X      (d) XLM-Indic HI-X

(e) Baseline BN-X      (f) XLM-Indic BN-X

(g) Baseline OR-X      (h) XLM-Indic OR-X

Figure 6: CKA of Baseline and XLM-Indic on all language pairs for pa, hi,bn and or

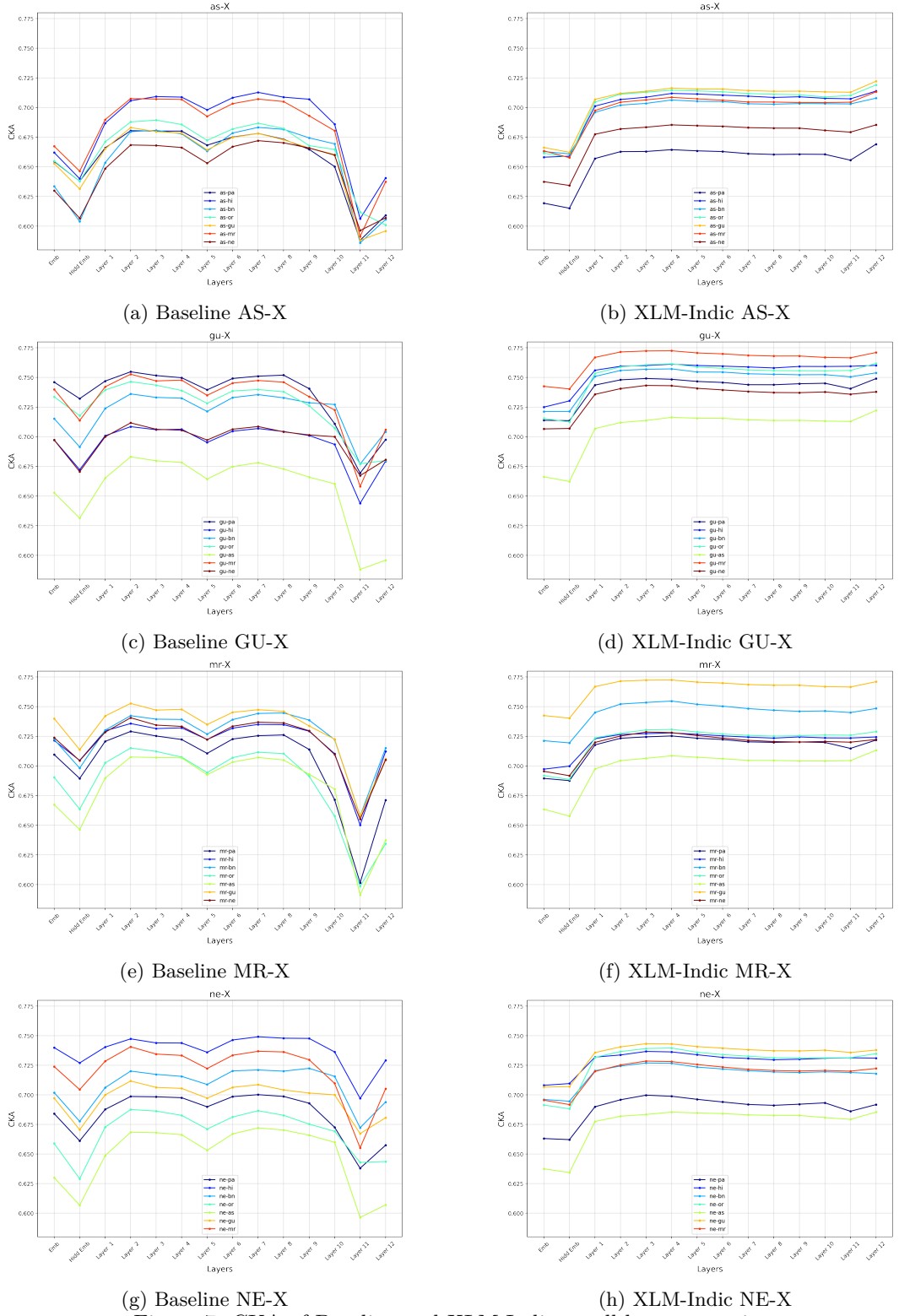

(a) Baseline AS-X            (b) XLM-Indic AS-X

(c) Baseline GU-X            (d) XLM-Indic GU-X

(e) Baseline MR-X            (f) XLM-Indic MR-X

(g) Baseline NE-X            (h) XLM-Indic NE-X

Figure 7: CKA of Baseline and XLM-Indic on all language pairs

