# OpenReview forum: "Transliteration: A Simple Technique For Improving Multilingual Language Modeling "
_ICLR.cc/2022/Conference — ICLR 2022 Submitted_

### Official Review · Reviewer_rME7 · 2021-10-30

**Correctness:** 3
**Technical Novelty And Significance:** 1
**Empirical Novelty And Significance:** 2
**Recommendation:** 5
**Confidence:** 4

**Main Review:**

Strength:
* The paper suggests a simple existing transliteration method (ISO-15919 standard) to convert Indic native characters into latin-based format. The transliterated format of a word tends to be shared across different Indic characters of different languages. This may have helped the model learn these languages better.
* The paper's proposal achieves good performance of multiple tasks in the South Asia languages.

Weaknesses:
* The paper considerably lacks novelty and does not meet the standard for ICLR. This method of transliteration already exists and have been adopted in other areas of NLP. The actual procedure of transliteration is also not invented by the authors, but taken from existing work. The paper simply suggests to convert text from one format to another by a known method, which I think is not novel.
* The method may not work for other languages, such as Chinese, where a transliteration from native Chinese character to Pinyin (probably) may not work well.
* Only Indic languages are explored. And I doubt this method would work in other languages with the same situation as South Asia languages, such as Chinese, Japanese, Korean; or other Asian languages like Thai, Malay, and Vietnamese.
* The presentation of the paper is quite poor. The method section is confusing, too much text, and difficult to comprehend and imagine. A simple diagram to aid understanding would make the presentation much better.



**Summary Of The Paper:**

The paper proposes to replace the native characters and words of some South Asia low-resource languages into latin-based transliteration using a rule-based converter and use these transliterated data to pre-train cross-lingual language model. The linguistic sharing of the transliterated format of these languages enable the model to alleviate out-of-vocabulary and limited data size to improve performance on the IndicGLUE benchmark, which is only focused on South Asia languages.

**Summary Of The Review:**

Overall, I think the paper significantly lacks novelty as only suggest an adoption of an existing method in a straightforward way that anyone would have thought of. The performance contribution is limited to a subset of languages and there is limited prospect that it would work in other languages and application.

**AFTER READING AUTHOR RESPONSE**

I have read the author response and very appreciate the authors' effort in responding. Here are my afterthoughts:
1. The novelty concern remained the same. The similar argument that "method X is applied in task A but never in B, so it's novel" has appeared frequently in past ICLR conferences. The general consensus is that this does not have sufficient technical novelty, as we value technical novelty in terms of machine learning techniques. It would have made a difference if the authors propose a novel technique Z based on X that makes things better for task B.
2. The authors acknowledge that this may only work in Indic languages, so it has limited application.
3. The paper, and the responses, show that the paper has lots of linguistic contribution. So I guess it will have better chance in core NLP venues, such as EMNLP.
4. I change the score to 5 to appreciate the authors' effort in the response and revision. However, this still does not warrant an acceptance for me.
5. The authors responded very late, on the last date of the deadline, which is not appreciative.

---

> ### Author Response · Authors · 2021-11-21
> **Response to rME7**
>
> **The paper considerably lacks novelty and does not meet the standard for ICLR. This method of transliteration already exists and have been adopted in other areas of NLP. The actual procedure of transliteration is also not invented by the authors, but taken from existing work. The paper simply suggests to convert text from one format to another by a known method, which I think is not novel.**
>
> We do agree that transliteration is not a new concept and has been studied somewhat well for Machine Translation. [1-5] show that transliteration can positively impact machine translation performance. However, to the best of our knowledge there has not been a rigorous empirical study on the impact of transliteration on the performance of multilingual language models. In fact, transliteration has not been used in any of the recent large multilingual language models such as mBERT, XLMRoberta, mT5. Muller [6] showed that for unseen languages, script barrier hindered transfer between low-resource and high-resource languages for multilingual language models and transliteration removed this barrier. They showed that transliterating Uyghur, Buryat, Erzya, Sorani, Meadow Mari, and Mingrelian to Latin script and finetuning mBERT on the respective corpus with masked language modeling objective improved their downstream POS performance significantly. [7] also found script barrier as a reason for mBERT and XLMRoberta’s poor performance on low resource languages. However, in our opinion, the key takeaway from our paper is that transliteration benefits the low-resource languages without negatively affecting the comparatively high-resource languages and we establish this finding through rigorous experiments and show the statistical significance along with the effect size of transliteration using the Mann-Whitney U test. . In the revised draft, we reported additional three different effect sizes besides p-value as p-value can be misleading [8]. The additional effect sizes were absolute effect size, common language effect size and standardized effect size. Also, to confirm that the uni-script model learns better representations we have added an analysis of the cross lingual similarity of the hidden representations of the models  in our revised draft. We apply centered kernel alignment (CKA) to measure cross-lingual representation similarity. We measure the CKA similarity score between the hidden representations of the models on the parallel sentences of eight Indo-Aryan languages from the FLORES-101 dataset. We find that, compared to the multi-script model, the uni-script model achieves a higher CKA score and it is more stable throughout the hidden layers of the uni-script model. Based on this, we conclude that the uni-script model learns better cross-lingual representation than the multi-script model.
>
> [1]. Haiyue Song, Raj Dabre, Zhuoyuan Mao, Fei Cheng, Sadao Kurohashi, and EiichiroSumita. Pre-training via leveraging assisting languages for neural machine translation. https://aclanthology.org/2020.acl-srw.37.
>
> [2] Vikrant Goyal, Sourav Kumar, and Dipti Misra Sharma. Eﬀicient neural machine trans-lation for low-resource languages via exploiting related languages. https://aclanthology.org/2020.acl-srw.22.
>
> [3]. Raj Dabre, Tetsuji Nakagawa, and Hideto Kazawa. An empirical study of language relatedness for transfer learning in neural machine translation. https://aclanthology.org/Y17-1038.
>
> [4] Chantal Amrhein and Rico Sennrich. On Romanization for model transfer between scripts inneural machine translation. https://aclanthology.org/2020.findings-emnlp.223
>
> [5] Yash Khemchandani, Sarvesh Mehtani, Vaidehi Patil, Abhijeet Awasthi, Partha Talukdar,and Sunita Sarawagi. Exploiting language relatedness for low web-resource language model adaptation: An Indic languages study. https://aclanthology.org/2021.acl-long.105
>
> [6] Benjamin Muller, Antonios Anastasopoulos, Benoît Sagot, and Djamé Seddah. When being unseen from mBERT is just the beginning: Handling new languages with multilingual language models.https://aclanthology.org/2021.naacl-main.38
>
> [7] Phillip Rust, Jonas Pfeiffer, Ivan Vulić, Sebastian Ruder, and Iryna Gurevych.  How good is your tokenizer?  on the monolingual performance of multilingual language models.  https://aclanthology.org/2021.acl-long.243
>
> [8] G. M. Sullivan and R. Feinn. Using Effect Size-or Why the P Value Is Not Enough.J GradMed Educ, 4(3):279–282, Sep 2012

---

> > ### Author Response · Authors · 2021-11-21
> > **Continuation of Response to rME7**
> >
> > **The method may not work for other languages, such as Chinese, where a transliteration from native Chinese character to Pinyin (probably) may not work well. Only Indic languages are explored. And I doubt this method would work in other languages with the same situation as South Asia languages, such as Chinese, Japanese, Korean; or other Asian languages like Thai, Malay, and Vietnamese.**
> >
> > We explore transliteration when closely related languages use different scripts as language relatedness is an important factor for multilingua language models. Chinese, Japanese, Korean, Thai, Malay and Vietnamese are from 6 different language families. In our paper we show that MLLM performance on closely related languages that use different scripts can benefit via transliteration through empirical and statistical analysis. We have not explored whether languages that are from different language families can benefit or get hurt from transliteration.
> > Restricting to Indo-Aryan has been a conscious choice on our part. Indo-Aryan family has one of the highest script diversity among different language families. Therefore, it presented the ideal scenario for script unification. In our revised draft, we have shown the script diversity encountered in our study in Table 5. We have also added additional studies in our Motivation and Background section (Section 2), to show the necessity of language relatedness for improvement of multilingual language modeling.  For example,[9] found that lexical overlap improved mBERT's multilingual representation capability even though it learned to capture multilingual representations with zero lexical overlaps. [10] showed that for unseen languages, script barrier hindered transfer between low-resource and high-resource languages for multilingual language models and transliteration removed this barrier. They showed that transliterating Uyghur, Buryat, Erzya, Sorani, Meadow Mari, and Mingrelian to Latin script and finetuning mBERT on the respective corpus with masked language modeling objective improved their downstream POS performance significantly. [11] also found script barrier as a reason for mBERT and XLMRoberta’s poor performance on low resource languages.
> >
> > **The presentation of the paper is quite poor. The method section is confusing, too much text, and difficult to comprehend and imagine. A simple diagram to aid understanding would make the presentation much better.**
> >
> > We thank the reviewer for their valuable comment. We have restructured and rewritten parts of the Experiment and Results (Section 3). Kindly take a look into our revised draft.
> >
> > [9] Telmo Pires, Eva Schlinger, and Dan Garrette. How multilingual is multilingual BERT? https://aclanthology.org/P19-1493
> >
> > [10] Benjamin Muller, Antonios Anastasopoulos, Benoît Sagot, and Djamé Seddah. When being unseen from mBERT is just the beginning: Handling new languages with multilingual language models.https://aclanthology.org/2021.naacl-main.38
> >
> > [11] Phillip Rust, Jonas Pfeiffer, Ivan Vulić, Sebastian Ruder, and Iryna Gurevych. How good is your tokenizer? on the monolingual performance of multilingual language models. https://aclanthology.org/2021.acl-long.243

---

> ### Author Response · Authors · 2021-11-27
> **Reply to rME7's Revised Response**
>
> We thank the reviewer for their reconsideration and valuable suggestion.
>
> We would like to clarify that we did not acknowledge that transliteration may only work in Indic languages. In our response we showed that past studies have demonstrated the efficacy of transliteration for related and unrelated languages. Specifically [1] and [2], where [1] shows the efficacy of transliteration for non-Indic languages on POS and [2] showed the efficacy of transliteration on non-Indic languages for NMT.
>
> Regarding the late response, the deadline of 22nd Nov 2021, was for the public comment and manuscript revision. The AC/Reviewer/Author discussion period is still ongoing and is in its final phase (23rd Nov to 29th Nov 2021). If the reviewer has any further questions, we would be happy to address their concerns.
>
> [1] Benjamin Muller, Antonios Anastasopoulos, Benoît Sagot, and Djamé Seddah. When being unseen from mBERT is just the beginning: Handling new languages with multilingual language models.https://aclanthology.org/2021.naacl-main.38
>
> [2] Chantal Amrhein and Rico Sennrich. On Romanization for model transfer between scripts in neural machine translation. https://aclanthology.org/2020.findings-emnlp.223

---

### Official Review · Reviewer_vmSB · 2021-11-01

**Correctness:** 3
**Technical Novelty And Significance:** 2
**Empirical Novelty And Significance:** 2
**Recommendation:** 3
**Confidence:** 5

**Main Review:**

Pros:
1. The idea of introducing transliteration into MLLM is interesting, where the ALBERT is pretrained secondarily on the OSCAR corpus.
2. The authors empirically validate their XLM-Indic under different settings. Besides the performance of downstream tasks after finetuning, results of zero-shot capability testing are also provided.
3. This paper also simply discusses why transliteration can improve the performance of MLLM by analyzing the subword fertility and unbroken ratio.

Cons:
1. One main concern about this paper is that the contribution is not enough. The purpose of this paper is to use the transliteration technique to improve the performance of MLLMs. However, only ALBERT is selected for validation (Other representative MLLMs such as mBERT, XLM are not discussed.). And there is no significant novelty of improving ALBERT to XLM-Indic such as better model architecture or pretraining objective (at least the improvement is not well presented in the paper).
2. The connection between transliteration technique and performance gain (on downstream tasks) is vague. Experimental results show that XLM-Indic (ALBERT size) significantly outperform existing MLLMs, while it is difficult for readers to get the intuition from the paper about how the performance gain is achieved, although Section 4 tries to provide some reasons (which are insufficient).
3. Many necessary details are missing. For example, the objective of pretraining ALBERT to XLM-Indic is not provided (only the hyper-parameters are provided in the appendix). There is no introduction about the details of XLM-Indic model. In addition, there is limited discussion about existing work. Using translation-based models to improve the performance of MLLMs is a widely-used strategy, and it would be better if authors can discuss and compare their XLM-Indic with existing methods.


Minor comments:
* In Tables 2&3, it would be nice to see the performance of downstream tasks on some common languages such as English. That would help readers to figure out whether knowledge of common languages is forgotten after the secondary pretraining.
* In Tables 5&6, the results about ablation study are a bit confusing. It would be better if a clearer description about the assumptions can be provided.


**Summary Of The Paper:**

This paper hypothesizes that transliterating all the languages to the same script can improve the performance of multilingual language models (MLLM). Experiments are designed to validate this hypothesis by comparing the performance of ALBERT and transliteration-based ALBERT (“XLM-Indic”). The authors find that XLM-Indic outperforms ALBERT across many downstream tasks such as text classification and QA. The main contribution is that the authors empirically validate the effectiveness of introducing transliteration technique on improving the overall performance of MLLMs. The improvement is obvious especially for underrepresented languages.

**Summary Of The Review:**

The paper does not provide a significant contribution and many necessary details are missing. It lacks the substance and depth to merit a strong recommendation.

---

> ### Author Response · Authors · 2021-11-21
> **Response to vmSB**
>
> Thank you for the detailed comments and suggestions!
>
> **One main concern about this paper is that the contribution is not enough. The purpose of this paper is to use the transliteration technique to improve the performance of MLLMs. However, only ALBERT is selected for validation (Other representative MLLMs such as mBERT, XLM are not discussed.). And there is no significant novelty of improving ALBERT to XLM-Indic such as better model architecture or pretraining objective (at least the improvement is not well presented in the paper)**
>
> We have not changed the architecture or pretraining objective of the ALBERT model. Our contribution is the empirical analysis of the effect of transliteration on MLLMs. We find that transliteration benefits the low-resource languages without negatively affecting the comparatively high-resource languages .We establish this finding through rigorous experiments and show the statistical significance along with the effect size of transliteration using the Mann-Whitney U test. In the current revised draft, we have added CKA analysis on section 4, to conclusively show that the uni-script model (XLM-Indic) learns better cross-lingual representations. We chose only ALBERT due to our design of experiment. Other studies have shown that transliteration helps MLLMs either by pretraining (XLM-Roberta) [1] or by adapting transliteration to their model (mBERT) [2].
>
> **The connection between transliteration technique and performance gain (on downstream tasks) is vague. Experimental results show that XLM-Indic (ALBERT size) significantly outperform existing MLLMs, while it is difficult for readers to get the intuition from the paper about how the performance gain is achieved, although Section 4 tries to provide some reasons (which are insufficient).**
>
> We disagree with the reviewer regarding their statement on the vagueness of connection between the performance gain and transliteration. Introduction of transliteration is the only difference between the multi-script model and uni-script model (XLM-Indic). We would like to clarify that the results of other papers were used as reference value only. We have revised our paper and added three different effect sizes along with statistical significance and have emphasized on our goal of measuring the effect of transliteration on MLLMs.  Additionally we have added CKA analysis on section 4, to conclusively show that the uni-script model (XLM-Indic) learns better cross-lingual representations than the multi-script model. Specifically, CKA score is higher and more stable for the uni-script model.
>
> **Many necessary details are missing. For example, the objective of pretraining ALBERT to XLM-Indic is not provided (only the hyper-parameters are provided in the appendix). There is no introduction about the details of XLM-Indic model. In addition, there is limited discussion about existing work. Using translation-based models to improve the performance of MLLMs is a widely-used strategy, and it would be better if authors can discuss and compare their XLM-Indic with existing methods.**
>
> We pretrain our models on Masked Language modeling and Sentence order prediction objectives. We have clarified this in the revised draft. We would be happy to provide any other details that the reviewer considers important regarding our XLM-Indic model. We do not think translation-based models are relevant for our study. We focus on unsupervised multilingual masked language models that learn from unaligned multilingual corpora such as mBERT, XLMRoberta, mT5 etc.
>
> **In Tables 2&3, it would be nice to see the performance of downstream tasks on some common languages such as English. That would help readers to figure out whether knowledge of common languages is forgotten after the secondary pretraining.**
>
> Our models are not pretrained on English corpus. We do not use the weight of the ALBERT model trained on English. We pretrain both the multis-cript model and the uni-script model on Indo Aryan languages **from scratch**. *There is no secondary pretraining.*
>
>
>
>
>
> [1] [Tejas Indulal Dhamecha, Rudra Murthy V, Samarth Bharadwaj, Karthik Sankaranarayanan, Pushpak Bhattacharyya. Role of Language Relatedness in Multilingual Fine-tuning of Language Models: A Case Study in Indo-Aryan Languages. EMNLP. 2021.] (https://aclanthology.org/2021.emnlp-main.675/)
>
> [2] [Muller, B., Anastasopoulos, A., Sagot, B., & Seddah, D. (2021, Junie). When Being Unseen from mBERT is just the Beginning: Handling New Languages With Multilingual Language Models. Proceedings of the 2021 Conference of the North American Chapter of the Association for Computational Linguistics: Human Language Technologies, 448–462. doi:10.18653/v1/2021.naacl-main.38] (https://aclanthology.org/2021.naacl-main.38/)

---

> > ### Author Response · Authors · 2021-11-21
> > **Continuation of Response to vmSB**
> >
> > **In Tables 5&6, the results about ablation study are a bit confusing. It would be better if a clearer description about the assumptions can be provided.**
> >
> > We have revised our draft and revised our assumption to only two tailed tests. We used Mann Whitney U test, which is a nonparametric hypothesis test between two groups/population. Under Mann Whitney U test, our null hypothesis is, both multi-script (group 2) and uni-script (group 1) models performance is equivalent. The alternative hypothesis is that the models’ performance are not equal.
> > In cases where we can reject the null hypothesis, we can confidently say that the uni-script model can outperform the multi-script model. Nevertheless, when we fail to reject the null hypothesis, we come to the conclusion that the models are likely equivalent. However, in our revised draft, we have also included three new test statistics to better interpret our results. We have included absolute effect size, common language effect size and standardized effect size. The absolute effect size δ is the difference between the mean of the models’ performance metric, which is given as,
> >
> > $\delta = \mu_{XLM-Indic}-\mu_{Baseline}$
> >
> > for any given task and language. For any given task, if the null hypothesis is rejected, a positive $\delta$ indicates the uni-script model is better, and a negative $\delta$ indicates the multi-script model is better. $\rho$ gives us the probability of superiority of one group being better performing between given two groups.That is the probability that a random performance sample of group 1 is greater than a random performance sample of group 2. The standardized effect size indicates the magnitude of difference between the performance values of group 1 and group 2. Standardized effect size shows us how realistically significant the performance differences are between models even if the performance difference is statistically significant.

---

### Official Review · Reviewer_M3CH · 2021-11-01

**Correctness:** 2
**Technical Novelty And Significance:** 1
**Empirical Novelty And Significance:** 2
**Recommendation:** 3
**Confidence:** 3

**Main Review:**

The main strength of this paper is that it provides a simple solution to the script suffering in language modeling, and compares its performance with several multilingual pretrained models as well as the non-transliteration version of the model. The transliteration ALBERT shows better performance over the non-transliterated version. The main weakness is that it only compares the performance within several Indian languages. And its claims may not stand in a larger scope. For example, this work tends to claim that tokens with the same spellings in different languages carry the same meanings. This claim is hard to stand. For instance, ‘burro’ has unrelated meanings in Spanish and Italian respectively, which are close languages under the same language family. And the lexical mapping may not be one-to-one even for different spelling systems of the same language like Simplified and Traditional Chinese (https://aclanthology.org/2020.acl-main.648.pdf). Additionally, it is not clear to see that shared tokens collapse to the same subword from the subword fertility analysis in Section 4. Finally, the experiments are not sufficient because script conversion and distinguishing closeness of languages are important problems in conducting this method. However, they are not fully discussed because the authors only select a group of very closed languages, and therefore cannot resolve the concerns when this technique is applied. In fact, since many languages can be Romanized, languages like Greek and Russian can be considered to test the hypothesis in this work.


**Summary Of The Paper:**

This paper talks about whether uniform script representation can alleviate the dominance of language-specific corpus size and therefore benefit cross-lingual language modeling. The authors train two ALBERTs on non-transliterated and transliterated (ISO-15919) corpus (filtered and normalized version of OSCAR). Those two models are further fine-tuned for four classification/sequence labeling tasks in IndicGLUE. The result shows that XLM-Indic achieves better or comparable performance in most settings. They also test the model’s zero shot capability in cloze style QA, and XLM-Indic also achieves better or comparable performance. By measuring lexical fertility, they claim that the success of shared scripts comes from that low-resource languages borrow better representation of shared lexical from other languages where those lexical are frequently used.

**Summary Of The Review:**

In summary, this work studies an important problem and tries to provide an effective solution. The experiment is done on a small scale and limited setting, and many underlying hypotheses of this work are not widely applied based on the insufficient results. I would not recommend its acceptance.

---

> ### Author Response · Authors · 2021-11-21
> **Response to M3CH**
>
> **The main weakness is that it only compares the performance within several Indian languages. And its claims may not stand in a larger scope.**
>
> Restricting to Indo-Aryan has been a conscious choice on our part. Indo-Aryan family has one of the highest script diversity among different language families. Therefore, it presented the ideal scenario for script unification. In our revised draft, we have shown the script diversity encountered in our study in Table 5. We have also added additional studies in our Motivation and Background section (Section 2), to show the necessity of language relatedness for improvement of multilingual language modeling.  For example, [1] found that lexical overlap improved mBERT's multilingual representation capability even though it learned to capture multilingual representations with zero lexical overlaps. [2] found that language relatedness is crucial for POS-tagging and dependency parsing tasks. Therefore, in order to measure the potential benefit of transliteration, it was necessary to restrict it within closely related languages.
>
> **For example, this work tends to claim that tokens with the same spellings in different languages carry the same meanings. This claim is hard to stand. For instance, ‘burro’ has unrelated meanings in Spanish and Italian respectively, which are close languages under the same language family.**
>
> Regarding words with the same spelling carrying the same meaning across languages, indeed words may or may not carry the same meaning. The reviewer has pointed out the example of the word ‘burro’ in Spanish and Italian carrying different meanings. However there are other word examples that do carry the same meaning. For example the word ‘libro’ means book in both Italian and spanish. The same is true for the word ‘casa’ in Spanish and italian. There are many such examples here (https://lingo-apps.com/same-words-spanish-italian/). Words such as ‘burro’ may indeed cause the issue of negative transfer while words such as ‘casa’ and ‘libro’ help the model by positive transfer. Ultimately this is a balancing act and whether this token sharing is beneficial or not must be determined based on downstream task performance.
>
> **And the lexical mapping may not be one-to-one even for different spelling systems of the same language like Simplified and Traditional Chinese (https://aclanthology.org/2020.acl-main.648.pdf).**
>
> We cannot comment on the particular merit of this transliteration scheme between Simplified and Traditional Chinese as we are not familiar with the Chinese writing system. However there are established ISO standards for transliteration of Greek (ISO 843), Cyrillic (ISO 9), Thai (ISO 11940), Indic (ISO 15919) etc scripts into Latin script. In these cases, our results suggest that transliteration may improve the performance of MLLMs.
>
> **Additionally, it is not clear to see that shared tokens collapse to the same subword from the subword fertility analysis in Section 4.**
>
> If there is a shared token collabs, we would expect the subword fertility to decrease and the unbroken ratio to increase as a result of that. Which is what happens as we have shown in section 4. This analysis is done as a form of sanity check. We have revised the wording on section 4 to clarify this. Kindly see our revised draft.
>
> [1] Phillip Rust, Jonas Pfeiffer, Ivan Vulić, Sebastian Ruder, and Iryna Gurevych. How good is your tokenizer? on the monolingual performance of multilingual language models. https://aclanthology.org/2021.acl-long.243
>
> [2] G. M. Sullivan and R. Feinn. Using Effect Size-or Why the P Value Is Not Enough.J GradMed Educ, 4(3):279–282, Sep 2012

---

> > ### Author Response · Authors · 2021-11-21
> > **Continuation of response to M3CH**
> >
> >
> > **Finally, the experiments are not sufficient because script conversion and distinguishing closeness of languages are important problems in conducting this method. However, they are not fully discussed because the authors only select a group of very closed languages, and therefore cannot resolve the concerns when this technique is applied. In fact, since many languages can be Romanized, languages like Greek and Russian can be considered to test the hypothesis in this work.**
> >
> > Distinguishing closeness of languages has already been done in different studies [3]. We agree that many languages  such as Greek and Russian can be romanized. Many languages written in cyrillic script have other closely related languages written in latin script (Turkic and South Slavic languages). However, [4] showed that transliteration helps improve performance even with unseen low-resource and unrelated languages. As stated earlier, script diversity was an important requirement for our design of experiment to measure the effect transliteration convincingly. There are other studies [5] that show that transliteration improves multilingual language performance and also adapting transliteration to existing models improves performance [3, 5]. Therefore, we would also like to restate that the goal of our study was to measure the effect of transliteration through rigorous empirical and statistical analysis. We have revised our draft to better reflect our emphasis on this statement along with the necessity of statistical testing.
> >
> > [3] Patrick Littell, David R. Mortensen, Ke Lin, Katherine Kairis, Carlisle Turner, and LoriLevin. URIEL and lang2vec: Representing languages as typological, geographical, and phylogenetic vectors. https://aclanthology.org/E17-2002
> >
> > [4] Benjamin Muller, Antonios Anastasopoulos, Benoît Sagot, and Djamé Seddah. When being unseen from mBERT is just the beginning: Handling new languages with multilingual language models.https://aclanthology.org/2021.naacl-main.38
> >
> > [5] Tejas Indulal Dhamecha, Rudra Murthy V, Samarth Bharadwaj, Karthik Sankaranarayanan, Pushpak Bhattacharyya. Role of Language Relatedness in Multilingual Fine-tuning of Language Models: A Case Study in Indo-Aryan Languages. EMNLP. 2021.
> >
> > [4] Yash Khemchandani, Sarvesh Mehtani, Vaidehi Patil, Abhijeet Awasthi, Partha Talukdar,and Sunita Sarawagi. Exploiting language relatedness for low web-resource language model adaptation: An Indic languages study. https://aclanthology.org/2021.acl-long.105

---

### Official Review · Reviewer_yNTm · 2021-11-02

**Correctness:** 3
**Technical Novelty And Significance:** 2
**Empirical Novelty And Significance:** 3
**Recommendation:** 6
**Confidence:** 4

**Main Review:**

**Strengths**

Through extensive experiments on a large number of tasks, the paper shows that a single script model is good for constrained scenarios when working with related languages:

- Small vocabulary (50k)
- high degree of parameter sharing (ALBERT)
- trained on limited corpora (OSCAR corpora)

A compact model trained on single script for related languages can be competitive/better than larger capacity models trained with multi-script data of a much larger magnitude. These findings are in line with concurrent work by Dhamecha et al (2021).

Tejas Indulal Dhamecha, Rudra Murthy V, Samarth Bharadwaj, Karthik Sankaranarayanan, Pushpak Bhattacharyya. Role of Language Relatedness in Multilingual Fine-tuning of Language Models: A Case Study in Indo-Aryan Languages. EMNLP. 2021.

**Weaknesses**

- Studies on the impact of vocabulary size, pre-training corpus size, and fine-tuning corpus size will be beneficial in understanding if transliteration is beneficial in constrained scenarios only. Some previous work indicates that large multilingual models can implicitly map-related languages into a common representation space even if the scripts are different.

   - Kudugunta, S. R., Bapna, A., Caswell, I., Arivazhagan, N., & Firat, O. (2019). Investigating multilingual NMT representations at scale. EMNLP.
   - Karthikeyan K, Zihan Wang, Stephen Mayhew, Dan Roth. Cross-Lingual Ability of Multilingual BERT: An Empirical Study. ICLR 2020.

**Questions**

- How are the hyperparameters selected? The same hyperparameters have been used for finetuning all models, is there a bias in the selection of these hyperparameters like early stopping, learning rate, etc for certain models? Kindly clarify.

**Comments**

- Please include citations and URLs for the IndicNLP and Aksharmukha library.
   - Kunchukuttan, Anoop. "The indicNLP library." Indian language NLP Library (2020).
   - https://github.com/virtualvinodh/aksharamukha-python



**Summary Of The Paper:**

In this paper, the authors show explore the following question: is it beneficial to represent different languages in the same script for training pre-training LMs. They explore this question for related languages which share similar scripts (making it easy to transliterate between scripts). The study has been performed on Indo-Aryan languages, which is an important representative of this scenario. The comparison between single-script and multi-script ALBERT language models show the single script model outperforms a multiscript model. The analysis shows that the single-script model vocabularies have a higher level of subword sharing.

**Summary Of The Review:**

The main contribution of the paper - that transliteration can help multilingual language models is an important observation for related languages. Though many in the community have believed this hypothesis, there were no solid empirical data to prove this for NLU.  Some work exists to show the benefits of single script mapping for NMT. Hence, this work fills that gap. Concurrent work by Dhamecha et al (2021) also confirms the major finding of the paper.

---

> ### Author Response · Authors · 2021-11-21
> **Response to yNTm**
>
> We thank the reviewer for their valuable comments.
>
> **These findings are in line with concurrent work by Dhamecha et al (2021)**
>
> We thank the reviewer for bringing to our attention the concurrent work of Dhamecha et al (2021). We have added it in our revised draft.
>
> **Please include citations and URLs for the IndicNLP and Aksharmukha library**
>
> We thank the reviewer for pointing out that we missed the IndicNLP library citation. We have added the citation on the revised draft. Aksharmukha has been cited since our earlier draft.
>
> **Studies on the impact of vocabulary size, pre-training corpus size, and fine-tuning corpus size will be beneficial in understanding if transliteration is beneficial in constrained scenarios only. Some previous work indicates that large multilingual models can implicitly map-related languages into a common representation space even if the scripts are different.**
>
> Indeed the usefulness of transliteration is somewhat well known for Machine Translation [1-5]. [6] has also shown that adapting transliteration to pretrained mBERT helps improve its performance for unseen languages. Regarding vocabulary size, we believe 50k is a reasonable number of tokens. For comparison, XLMRoberta which is pretrained on about 100 languages uses 250k tokens, mBERT trained on 104 languages uses about 120k tokens, while our models pretrained on 14 languages use 50k tokens. Regarding fine tuning corpus size, the News Classification Task IndicGLUE presents us with the opportunity to test this. We reported this in our earlier draft. However we have revised our draft to discuss this more clearly. Kindly take a look at the revised draft.
>
> Regarding [7], we believe Machine Translation is crucially different from Unsupervised MLLM in that the former uses the supervision of parallel corpus. When parallel corpus is available the model can use that information to align the embeddings of two languages that use different scripts. In this paper we deal with the scenario of training a multilingual model in the absence of parallel corpora. Regarding the same paper we believe the proper way to study the impact of scripts would be to train two NMT models, one where we use transliteration and one where we keep the original scripts and measure the difference between the performance of the two models. This is exactly what we do in the context of MLLM.
>
> Regarding [8], we do agree that mBERT can learn some cross-lingual representations without lexical overlap. In this paper we study whether models can learn better cross-lingual representations when we apply transliteration.
>
>
> [1]. Haiyue Song, Raj Dabre, Zhuoyuan Mao, Fei Cheng, Sadao Kurohashi, and EiichiroSumita.  Pre-training via leveraging assisting languages for neural machine translation. https://aclanthology.org/2020.acl-srw.37.
>
> [2] Vikrant Goyal, Sourav Kumar, and Dipti Misra Sharma. Eﬀicient neural machine trans-lation for low-resource languages via exploiting related languages. https://aclanthology.org/2020.acl-srw.22.
>
> [3]. Raj Dabre, Tetsuji Nakagawa, and Hideto Kazawa. An empirical study of language relatedness for transfer learning in neural machine translation. https://aclanthology.org/Y17-1038.
>
> [4] Chantal Amrhein and Rico Sennrich. On Romanization for model transfer between scripts inneural machine translation. https://aclanthology.org/2020.findings-emnlp.223
>
> [5] Yash Khemchandani, Sarvesh Mehtani, Vaidehi Patil, Abhijeet Awasthi, Partha Talukdar,and Sunita Sarawagi. Exploiting language relatedness for low web-resource language model adaptation: An Indic languages study. https://aclanthology.org/2021.acl-long.105
>
> [6] Benjamin Muller, Antonios Anastasopoulos, Benoît Sagot, and Djamé Seddah. When being unseen from mBERT is just the beginning: Handling new languages with multilingual language models.https://aclanthology.org/2021.naacl-main.38
>
> [7] Kudugunta, S. R., Bapna, A., Caswell, I., Arivazhagan, N., & Firat, O. (2019). Investigating multilingual NMT representations at scale. EMNLP.
>
> [8] Karthikeyan K, Zihan Wang, Stephen Mayhew, Dan Roth. Cross-Lingual Ability of Multilingual BERT: An Empirical Study. ICLR 2020.

---

> > ### Author Response · Authors · 2021-11-21
> > **Continuation of Response to yNTm**
> >
> > **How are the hyperparameters selected? The same hyperparameters have been used for finetuning all models, is there a bias in the selection of these hyperparameters like early stopping, learning rate, etc for certain models? Kindly clarify.**
> >
> > We report the hyperparameters in the appendix (A.4). We use the same hyperparameter for fine tuning the multi-script and the uni-script model. We haven't used early stopping in finetuning the models.  The batch size was chosen to be the maximum that fits in memory. This was done so that each batch contains approximately the same number of tokens. Otherwise the hyperparameters were chosen following the recommendations of [9]. On the highly skewed IITP Movie Review, IITP Product Review and MIDAS Discourse we found that this default setting resulted in worse performance compared to the independent baselines. So we finetuned the learning rate and classifier dropout on the validation set of these tasks. On the comparatively large datasets, Wikipedia Section Title Prediction and Soham News Classification we use 3 epochs and 8 epochs respectively. These were chosen to limit the fine-tuning execution time to at most 8 hours.
> >
> > [9] Marius Mosbach, Maksym Andriushchenko, and Dietrich Klakow. On the stability of finetuning BERT: misconceptions, explanations, and strong baselines. OpenReview.net, 2021.https://openreview.net/forum?id=nzpLWnVAyah.

---

### Official Review · Reviewer_XxGN · 2021-11-03

**Correctness:** 3
**Technical Novelty And Significance:** 2
**Empirical Novelty And Significance:** 2
**Recommendation:** 5
**Confidence:** 4

**Main Review:**

Strengths -
1) The paper is well written and easy to follow.
2) Strong results with a thorough study on many languages and also statistical tests.
3) A simple yet effective way to increase the sharing of information across languages.

Weaknesses -
1) Transliteration is not formally defined, which is leading to authors not making a connection with the works that do the same thing but by representing them into phonemes.
2) Additionally, this leads to being restricted to only the Indo-Aryan languages and not expanding to a much larger set of languages around the world, by using various available g2p tools.
3) The paper seems to have also missed a lot of research in this direction which do almost exactly the same thing, for cross-lingual downstream NLP tasks or infact multilingual language modeling itself. I have listed a few of them as part of the general remarks.
4) The core-idea and the contributions made by this paper is not very novel. It's been used a lot in the field of speech and NLP. Although the authors apply this work to a new setup and on a new corpora, the key takeaways are not new or surprising.

General Remarks -
1) In section 3.2 : “As our batch size is 1/16th of the ALBERT paper, we use a learning rate of 1e-3/8, which is approximately 1/16th of the learning rate used in the ALBERT paper (1.76e-2).”  You could instead be doing accumulation of gradients instead of change the learning rates. By doing that you can increase your effective batchsize with limited compute.
2) From your code it seems that you used hugging face implementation for the model training and tuning and evaluation? It would be good to mention that in the paper.
3) “However, we would like to mention that, in this paper our purpose is not to achieve the SOTA but to understand the impact of transliteration on the performance of language models.”   → the paper is mentioning SOTA and results everywhere (even half of the abstract) and is studying very quickly transliteration in the last section (section 4) in about 15 lines and one figure which is not very striking and not very commented
4) The authors mention "However, we would like to mention that, in this paper our purpose is not to achieve the SOTA but to understand the impact of transliteration on the performance of language models.”. But the paper does seem to be focused almost entirely on results with the final section on studying the transliteration.
5) Expanding to this point, it would be good to look at unrelated languages and get a sense of the subword fertility in both related cluster and unrelated cluster of languages.
6) In section 3.4 - the authors mentioned that they modified the evaluation code. I always wary on the side of caution about editing evaluations. It would be good to get it verified by the IndicBERT github codebase.
7) It would probably be a good idea to make a relation with the works that use phoneme representations for language modeling and downstream speech and NLP tasks. I have listed a few that come to my mind below -


[a] Phonologically Aware Neural Model for Named Entity Recognition in Low Resource Transfer Settings, EMNLP 2016

[b] Polyglot Neural Language Models: A Case Study in Cross-Lingual Phonetic Representation Learning, NAACL 2016

[c] Zero-shot Neural Transfer for Cross-lingual Entity Linking, AAAI 2019

[d] Phoneme Level Language Models for Sequence Based Low Resource ASR, ICASSP 2019

[e] Exploring Phoneme-Level Speech Representations for End-to-End Speech Translation, ACL 2019

[f] On Romanization for Model Transfer Between Scripts in Neural Machine Translation, EMNLP 2020

[g] Using Phoneme Representations to Build Predictive Models Robust to ASR Errors, ACM SIGIR 2020

[h] Phonotactic Complexity and Its Trade-offs, TACL 2020


**Summary Of The Paper:**

This paper uses transliteration to build better multilingual language models. This is particularly useful for building downstream NLP models for low resource languages. The idea behind this is that script divergences between languages avoid pretrained models to pool resources across language, leading to poor performances on these scripted low-resource languages. Transliteration, i.e. using a common script for all the languages can help solve this issue.

For experiments, the authors focus on Indo-Aryan languages, where they first transliterate all the documents to a common script. Then, the ALBERT model is pre-trained on the transliterated corpora of these languages extracted from the OSCAR corpus. For downstream tasks (section title prediction, news category classification, NER and genre/sentiment classification), they perform fine tuning for each tasks independently, in a multilingual manner when possible. The transliterated models outperform the baseline systems (with no transliteration) on most tasks.


**Summary Of The Review:**

The authors perform extensive experimentations on many languages and perform statistical tests by training many models with different random seeds. The paper is also well written and easy to understand. However, there are several limitations in the scope and relation to prior work. Few major limitation in my opinion are - (1) Not relating transliteration to grapheme to phonemes. In fact this realization would have encouraged the authors to expand from Indo-Aryan languages to a much larger set of languages. Additionally, this would have also have led to nice set of relation to prior work. (2) Although it's important to consider internationalization of language technologies, the core-idea and the contributions made by this paper is not really novel. A lot of prior work have already shown the efficacy of this direction of research.

**After the Author Response** --
Thank you for the detailed response and the changes in the paper. The paper has indeed improved quite a bit from its previous shape. As the other reviewers also pointed out, I am still concerned about the novelty of the work so I am inclined to keep my score the same. However, with the new changes I think now the paper brings some insights that could be useful to some readers in the community. I would highly encourage the authors to submit this paper in a core NLP conference. Finally, please refrain from making significant edits between revisions as it really increases the burden on the reviewers.

---

> ### Author Response · Authors · 2021-11-21
> **Response to XxGN**
>
> **The paper is mentioning SOTA and results everywhere (even half of the abstract) and is studying very quickly transliteration in the last section (section 4) in about 15 lines and one figure which is not very striking and not very commented.**
>
> We thank the reviewer for this valuable suggestion. We have removed mentions of SOTA from our revised draft and emphasized more on statistical analysis. We have also added additional analysis on transliteration in Section 4. We applied centered kernel alignment to show the comparison between multi-script and uni-script models.
>
> **The core-idea and the contributions made by this paper is not very novel. It's been used a lot in the field of speech and NLP. Although the authors apply this work to a new setup and on a new corpora, the key takeaways are not new or surprising.**
>
> We do agree that transliteration is not a new concept and has been studied somewhat well for Machine Translation. [1-5] show that transliteration can positively impact machine translation performance. However, to the best of our knowledge there has not been a rigorous empirical study on the impact of transliteration on the performance of multilingual language models. In fact, transliteration has not been used in any of the recent large multilingual language models such as mBERT, XLMRoberta, mT5. [6] showed that for unseen languages, script barrier hindered transfer between low-resource and high-resource languages for multilingual language models and transliteration removed this barrier. They showed that transliterating Uyghur, Buryat, Erzya, Sorani, Meadow Mari, and Mingrelian to Latin script and finetuning mBERT on the respective corpus with masked language modeling objective improved their downstream POS performance significantly. [7] also found script barrier as a reason for mBERT and XLMRoberta’s poor performance on low resource languages.
>
> However, in our opinion, the key takeaway from our paper is that transliteration benefits the low-resource languages without negatively affecting the comparatively high-resource languages and we establish this finding through rigorous experiments and show the statistical significance along with the effect size of transliteration using the Mann-Whitney U test. . In the revised draft, we reported additional three different effect sizes besides p-value as p-value can be misleading [8]. The additional effect sizes were absolute effect size, common language effect size and standardized effect size. Also, to confirm that the uni-script model learns better representations we have added an analysis of the cross lingual similarity of the hidden representations of the models  in our revised draft. We apply centered kernel alignment (CKA) to measure cross-lingual representation similarity. We measure the CKA similarity score between the hidden representations of the models on the parallel sentences of eight Indo-Aryan languages from the FLORES-101 dataset. We find that, compared to the multi-script model, the uni-script model achieves a higher CKA score and it is more stable throughout the hidden layers of the uni-script model. Based on this, we conclude that the uni-script model learns better cross-lingual representation than the multi-script model.
>
> [1]. Haiyue Song, Raj Dabre, Zhuoyuan Mao, Fei Cheng, Sadao Kurohashi, and EiichiroSumita. Pre-training via leveraging assisting languages for neural machine translation. https://aclanthology.org/2020.acl-srw.37.
>
> [2] Vikrant Goyal, Sourav Kumar, and Dipti Misra Sharma. Eﬀicient neural machine trans-lation for low-resource languages via exploiting related languages. https://aclanthology.org/2020.acl-srw.22.
>
> [3]. Raj Dabre, Tetsuji Nakagawa, and Hideto Kazawa. An empirical study of language relatedness for transfer learning in neural machine translation. https://aclanthology.org/Y17-1038.
>
> [4] Chantal Amrhein and Rico Sennrich. On Romanization for model transfer between scripts inneural machine translation. https://aclanthology.org/2020.findings-emnlp.223
>
> [5] Yash Khemchandani, Sarvesh Mehtani, Vaidehi Patil, Abhijeet Awasthi, Partha Talukdar,and Sunita Sarawagi. Exploiting language relatedness for low web-resource language model adaptation: An Indic languages study. https://aclanthology.org/2021.acl-long.105
>
> [6] Benjamin Muller, Antonios Anastasopoulos, Benoît Sagot, and Djamé Seddah. When being unseen from mBERT is just the beginning: Handling new languages with multilingual language models.https://aclanthology.org/2021.naacl-main.38
>
> [7] Phillip Rust, Jonas Pfeiffer, Ivan Vulić, Sebastian Ruder, and Iryna Gurevych.  How good is your tokenizer?  on the monolingual performance of multilingual language models.  https://aclanthology.org/2021.acl-long.243
>
> [8] G. M. Sullivan and R. Feinn. Using Effect Size-or Why the P Value Is Not Enough.J GradMed Educ, 4(3):279–282, Sep 2012

---

> > ### Author Response · Authors · 2021-11-21
> > **Continuation of Response to XxGN**
> >
> > **The paper seems to have also missed a lot of research in this direction which do almost exactly the same thing, for cross-lingual downstream NLP tasks or infact multilingual language modeling itself. I have listed a few of them as part of the general remarks.**
> >
> > We would like to thank the reviewer for their thorough suggestions. We do agree that our first draft had shortcomings regarding phoneme studies. In our revised draft we have revised the Motivation and Background section (Section 2) and included the referred studies [a],[b] and [c]. However, we previously did cite [f] in our earlier draft in Section 2.
> >
> > **Additionally, this leads to being restricted to only the Indo-Aryan languages and not expanding to a much larger set of languages around the world, by using various available g2p tools.**
> >
> > Restricting to Indo-Aryan has been a conscious choice on our part. Indo-Aryan family has one of the highest script diversity among different language families. Therefore, it presented the ideal scenario for script unification. In our revised draft, we have shown the script diversity encountered in our study in Table 5. We have also added additional studies in our Motivation and Background section (Section 2), to show the necessity of language relatedness for improvement of multilingual language modeling.  For example, [9] found that lexical overlap improved mBERT's multilingual representation capability even though it learned to capture multilingual representations with zero lexical overlaps. [10] found that language relatedness is crucial for POS-tagging and dependency parsing tasks. Therefore, in order to measure the potential benefit of transliteration, it was necessary to restrict it within closely related languages.
> >
> > **Transliteration is not formally defined, which is leading to authors not making a connection with the works that do the same thing but by representing them into phonemes.**
> >
> > We used an ISO standard (ISO 15919) for our transliteration scheme. Including phonemes in our studies due to the scope of it is not possible. Due to our design of experiment, adding phoneme would have increased the complexity of both our ablation study and statistical analysis. For this specific reason, during our revision, we kept the scope specifically on transliteration and did not explore g2p. In their paper, [11] showed that both g2p and romanization improved cross-lingual morphological inflection performance. They only showed statistical significance and did not report effect size. However, from their study, the comparison between g2p and romanization is not conclusive. Therefore, there is a scope for comparing the individual effects of g2p and romanization.
> >
> > **In section 3.2 : “As our batch size is 1/16th of the ALBERT paper, we use a learning rate of 1e-3/8, which is approximately 1/16th of the learning rate used in the ALBERT paper (1.76e-2).” You could instead be doing accumulation of gradients instead of change the learning rates. By doing that you can increase your effective batchsize with limited compute.**
> >
> > We did consider gradient accumulation and initially tried to use it. However, our training time increased proportionally to the number of accumulated batches. Given that in our current setting we pre-trained for about one week, simulating large batch size via gradient accumulation would have been too costly,  given our limited computational budget.
> >
> > **From your code it seems that you used hugging face implementation for the model training and tuning and evaluation? It would be good to mention that in the paper.**
> >
> > We thank the reviewer for bringing this to our attention. This was an oversight on our part. We have added the citation in the revised draft.
> >
> > **Expanding to this point, it would be good to look at unrelated languages and get a sense of the subword fertility in both related cluster and unrelated cluster of languages.**
> >
> > We are not sure what the reviewer means by related cluster and unrelated cluster languages. We would be happy to perform additional experiments if the reviewer clarifies this point.
> >
> > [9] Telmo Pires, Eva Schlinger, and Dan Garrette. How multilingual is multilingual BERT? https://aclanthology.org/P19-1493
> >
> > [10] Anne Lauscher, Vinit Ravishankar, Ivan Vulić, and Goran Glavaš. From zero to hero: Onthe limitations of zero-shot language transfer with multilingual Transformers. https://aclanthology.org/2020.emnlp-main.363
> >
> > [11] N. Murikinati, A. Anastasopoulos, en G. Neubig, Transliteration for Cross-Lingual Morphological Inflection. https://aclanthology.org/2020.sigmorphon-1.22/

---

> > > ### Author Response · Authors · 2021-11-21
> > > **Continuation of Response to XxGN**
> > >
> > > **In section 3.4 - the authors mentioned that they modified the evaluation code. I always wary on the side of caution about editing evaluations. It would be good to get it verified by the IndicBERT github codebase.**
> > >
> > > On the appendix of the revised draft, we have added a discussion in this regard. The issue with IndicBERT code is that it does not take into account subword tokenization of a word into multiple tokens. We account for this in our evaluation. Since, our goal is not to establish SOTA but rigorously study the effect of transliteration, we believe the results reported with our evaluation method (taking into account effect of subword tokenization) is more appropriate. On the other hand, if we were to establish SOTA results, then using the same (possibly flawed) evaluation method as the other papers would have been necessary.
> > >
> > > **It would probably be a good idea to make a relation with the works that use phoneme representations for language modeling and downstream speech and NLP tasks. I have listed a few that come to my mind below -**
> > >
> > > We thank the reviewer for their suggestion. Our Motivation and Background section (Section 2) has been updated in the revised draft and we have included [a],[b] and [c] as we felt those fit in our scope well. However, we would like to mention that [f] was present in the previous draft in Section 2.

---

> ### Author Response · Authors · 2021-11-28
> **Reply to XxGN's Revised Response**
>
> We thank the reviewer for their suggestion. Nevertheless, it is discouraging that the score remained the same due to novelty only, even after significant improvement to the paper as recognized by the reviewer.

---

### Author Response · Authors · 2021-11-21
**Summary of Revisions**

**Abstract**

-Revised Abstract

**Introduction**

-Revised Introduction

**Motivation and Background**

-Added studies on language relatedness

-Added studies on script barrier issues

-Added studies on phoneme based methods

-Expanded upon previous background

-Removed Table 1 from previous draft

-Moved details on linguistic relatedness of indo-aryan family to appendix A.1

**Experiments and Results**

-Moved pretraining model hyperparameter details to appendix

-Added three new test statistics under Statistical Analysis section

-Removed Table 5 and 6 of previous draft

-Restructured result tables

-Added subsubsection on Effects of Transliteration

**Why Transliteration works**

-We have rewritten this section to clarify the motivation behind the subword fertility analysis.

-We have added an analysis of cross-lingual similarity of the hidden state representations of the models to determine whether transliteration to a common script helps MLLM learn better cross-lingual representations.

**Appendix**

-Added cross-lingual similarity plot among all language pairs.

-Added pretraining dataset statistics

-Added discussion regarding our choice of downstream hyperparameters

-Added public dataset test accuracy results table

---

### Author Response · Authors · 2021-11-21
**General Response to All Reviewers**

**Recap: What is our goal?**

Our primary goal is to measure the effect of transliteration on multilingual language models.

**Recap: What is not the goal?**

The goal is not to establish a new SOTA based on transliteration on the IndicGLUE benchmark.

**Recap: Why this goal?**

A large body of work suggests that language-relatedness can help MLLMs achieve better performance on low-resource languages by leveraging related high-resource languages. However, one of the major challenges in leveraging transfer between closely related high-resource and low-resource languages is overcoming the script barrier.. [1] showed that transfer learning in the same or linguistically similar language family gives the best performance for NMT. [2] studied how transliteration improved NMT and came to the conclusion that transliteration offered significant improvement for low-resource languages with different scripts.
[3] found that language relatedness is crucial for POS-tagging and dependency parsing tasks. [4] found that lexical overlap improved mBERT's multilingual representation capability.[5] showed on Indo-Aryan languages that language relatedness could be exploited through transliteration along with bilingual lexicon-based pseudo-translation and aligned loss to incorporate low-resource languages into pretrained mBERT. [6] showed that for unseen languages, script barrier hindered transfer between low-resource and high-resource languages for MLLMs and transliteration removed this barrier. [7] showed that transliterating Indo-Aryan languages improved multilingual language model performance. From the literature, it can be seen that many in the community believe transliteration to be a potential solution for script barriers. However, most of the work shows the benefits of transliteration for NMT. Nevertheless, there is no solid empirical analysis of the effects of transliteration for MLLMs.
The goal of this paper is to provide a solid empirical analysis of the effect of transliteration for MLLMs with statistical analysis and determine if it helps models learn better cross-lingual representation.

**Recap: How did we achieve this goal?**

We empirically measure the effect of transliteration on the performance of MLLMs by focusing on the Indo-Aryan language family. We pretrain two ALBERT models from scratch, where one is pretrained with original scripts and the other after transliterating to a common script. Afterward, we evaluate the models on the IndicGLUE benchmark. We used Mann Whitney U test to determine whether the effect of transliteration is significant by analyzing the p-value, absolute effect size, common language effect size and standardized effect size. We also measure the cross-lingual representation similarity of the models using centered kernel alignment on parallel sentences of eight languages from the FLORES-101 dataset.

**Recap: What are our findings?**
We find that transliteration benefits the low-resource languages without negatively affecting the comparatively high-resource languages .We establish this finding through rigorous experiments and show the statistical significance along with the effect size of transliteration using the Mann-Whitney U test. We also find that different effect sizes are useful for better model comparison. Using CKA on the FLORES-101 dataset, we show that transliteration helps MLLMs learn better cross-lingual representation.

[1]. Raj Dabre, Tetsuji Nakagawa, and Hideto Kazawa. An empirical study of language relatedness for transfer learning in neural machine translation. https://aclanthology.org/Y17-1038.

[2] Chantal Amrhein and Rico Sennrich. On Romanization for model transfer between scripts inneural machine translation. https://aclanthology.org/2020.findings-emnlp.223

[3] Anne Lauscher, Vinit Ravishankar, Ivan Vulić, and Goran Glavaš. From zero to hero: Onthe limitations of zero-shot language transfer with multilingual Transformers. https://aclanthology.org/2020.emnlp-main.363

[4] Telmo Pires, Eva Schlinger, and Dan Garrette. How multilingual is multilingual BERT? https://aclanthology.org/P19-1493

[5] Yash Khemchandani, Sarvesh Mehtani, Vaidehi Patil, Abhijeet Awasthi, Partha Talukdar,and Sunita Sarawagi. Exploiting language relatedness for low web-resource language model adaptation: An Indic languages study. https://aclanthology.org/2021.acl-long.105

[6] Benjamin Muller, Antonios Anastasopoulos, Benoît Sagot, and Djamé Seddah. When being unseen from mBERT is just the beginning: Handling new languages with multilingual language models.https://aclanthology.org/2021.naacl-main.38

[7] Tejas Indulal Dhamecha, Rudra Murthy V, Samarth Bharadwaj, Karthik Sankaranarayanan, Pushpak Bhattacharyya. Role of Language Relatedness in Multilingual Fine-tuning of Language Models: A Case Study in Indo-Aryan Languages. EMNLP. 2021.

---

> ### Author Response · Authors · 2021-11-21
> **Continuation of General Response to All Reviewers**
>
> **Recap: What are the common concerns among reviewers and how do we address them?**
>
> **Novelty:** A common concern between reviewers **XxGN**, **M3CH**, **vmSB** and **rME7** was that the study was not novel enough. However, our goal was to empirically measure how effective transliteration is for multilingual language models. Our earlier draft failed to highlight this and we believe this is what led to the concerns regarding novelty. Therefore, keeping the reviewers’ concern in mind, we have revised our manuscript and have put more emphasis on rigorously measuring effectiveness of transliteration.
> In our revision we expanded upon our statistical analysis and revised our assumption to only two tailed tests. We used Mann Whitney U test, which is a nonparametric hypothesis test between two groups/population. Under Mann Whitney U test, our null hypothesis was, both multi-script (group 2) and uni-script (group 1) models performance is equivalent. The alternative hypothesis was that the models’ performance are not equal. In cases where we could reject the null hypothesis, our results showed that the uni-script model outperformed the multi-script model. Nevertheless, when we failed to reject the null hypothesis, we came to the conclusion that the models are likely equivalent. However, in our revised draft, we have also included three new test statistics to better interpret our results. We have included absolute effect size, common language effect size and standardized effect size. The absolute effect size $\delta$ is the difference between the mean of the models’ performance metric, which is given as,
>
> $\delta=\mu_{uni-script}-\mu_{multi-script}$
>
> for any given task and language. For any given task, when the null hypothesis is rejected, we can conclude that the difference $\delta$ is statistically significant. In this case, a positive $\delta$ indicates the uni-script model is better, and a negative $\delta$ indicates the multi-script model is better. $\rho$ gives us the probability of superiority of one group being better performing between given two groups. That is the probability that a random performance sample of group 1 is greater than a random performance sample of group 2. The standardized effect size indicates the magnitude of difference between the performance values of group 1 and group 2. Standardized effect size shows us how realistically significant the performance differences are between models even if the performance difference is statistically significant.
>
> In our revised draft, we have added an analysis of the cross lingual similarity of the hidden representations of the models. We apply centered kernel alignment (CKA) to measure cross-lingual representation similarity. We measure the CKA similarity score between the hidden representations of the models on the parallel sentences of eight Indo-Aryan languages from the FLORES-101 dataset. We find that, compared to the multi-script model, the uni-script model achieves a higher CKA score and it is more stable throughout the hidden layers of the uni-script model.Based on this, we conclude that the uni-script model learns better cross-lingual representation than the multi-script model.
> Finally, in the revised draft, we also expanded our background work and motivation to include previously missed studies.

---

> > ### Author Response · Authors · 2021-11-21
> > **Continuation of General Response to All Reviewers**
> >
> > **Small number of languages:** Reviewer **rME7**, **XxGN** and **M3CH** raised the concern that our results are restricted to a small set of languages and may not generalize to other languages. Our choice to restrict the analysis of Indo-Aryan languages is three-fold.
> > First, to isolate the potential benefit brought by transliteration by increasing lexical overlap among closely related languages, from the negative impact of lexical overlap with unrelated languages, we need to restrict our models to only closely related languages. As found by [1,2,3], language relatedness is also an important factor when it comes to cross-lingual transfer between low-resource to high resource language.
> > Second, the Indo-Aryan language family has the highest script diversity. Among the languages we pretrain our model on, we find six different scripts. On table 5 of our appendix we list the different scripts used by the languages. Thus this language family presents a practical scenario where the issue of script diversity among closely related languages is present.
> > Third, to rigorously measure the effect of transliteration we need a diverse set of benchmark tasks. The IndicGLUE benchmark dataset provides diverse benchmark tasks on seven languages of the IndoAryan family. The availability of the IndicGLUE benchmark dataset is a major reason for choosing these languages for our analysis.
> >
> > **Shortcomings of background work:** Reviewer **XxGn** showed concern regarding our background work in our earlier draft for missing phoneme based studies. Reviewer **yNTm** suggested us three papers of which one was our contemporary study.  To resolve these issues, we expanded on our background section  and included studies on the necessity of language relatedness for multilingual language models, drawbacks of script barrier, grapheme to phoneme based methods and use of transliteration for multilingual language modeling. Reviewer **XxGn** recommended us eight papers. Out of those eight, one was already cited in the previous draft and in the revised draft we have added three relevant papers out of the eight suggested papers. We have also added two papers from reviewer **yNTm**’s recommendations.
> >
> > [1] Telmo Pires, Eva Schlinger, and Dan Garrette. How multilingual is multilingual BERT? https://aclanthology.org/P19-1493
> >
> > [2] Anne Lauscher, Vinit Ravishankar, Ivan Vulić, and Goran Glavaš. From zero to hero: Onthe limitations of zero-shot language transfer with multilingual Transformers. https://aclanthology.org/2020.emnlp-main.363
> >
> > [3] Yash Khemchandani, Sarvesh Mehtani, Vaidehi Patil, Abhijeet Awasthi, Partha Talukdar,and Sunita Sarawagi. Exploiting language relatedness for low web-resource language model adaptation: An Indic languages study. https://aclanthology.org/2021.acl-long.105
> >
> > ## Finally, we are thankful to all the reviewers for their insightful comments and suggestions.

---

### Decision · Program_Chairs · 2022-01-20

**Decision:**

Reject

**Comment:**

The authors show that it is possible to overcome the script barrier in MLLMs by using transliteration. In effect, they show that transliterating all text to a single script improves the performance for low-resource languages. They also provide additional analysis in the form of statistical tests and crosslingual representation analysis to substantiate their claims.

The main concerns raised by the reviewers are:
(i) lack of novelty: the idea of using transliteration has been extensively studied in the context of NMT, Speech. It has also been studied in the context of MLLMs by some recent work (which can be considered to be contemporary). IMO, this is a concern.
(ii) focus on Indic languages: there are some concerns raised about the broader applicability of the techniques presented in the paper (personally, I disagree with this concern as Indic languages are important - for example, there are numerous papers which only report results on En-De, En-Ru translation)
(iii) limited evaluation: the technique is evaluated only using the ALBERT model and other configurations (such as ROBERTA, XLM, etc) are not considered. IMO, it would have helped if the authors presented results on these models also (at least we would know if transliteration only helps in the case of small/compact models or even in the case of large models)
(iv) missing references: there is a large body of related work on NMT, speech, etch which the authors had missed in their initial draft. This has been rectified in the updated version.

The reviewers did participate in the discussion with the meta-reviewer (not with the authors though) and even after looking at the revised draft mentioned that the novelty is limited.

To summarise my views, I think the initial draft of the paper did need improvements and the final draft is a significantly improved version of the initial draft. However, I still feel the novelty is missing. Even the empirical novelty claimed by the authors is ;lacking due to the use of a single model (ALBERT).